# Cooperative pathways for the green transformation of heavily polluting enterprises: A four-party game-driven mechanism for green M&A

**Mengxin Sun** 🔍*◉, **Xianggang Huang**◉

School of Economics and Management, China University of Mining and Technology, Xuzhou, China

◉ These authors contributed equally to this work.
* sunmengxin98@163.com

## Abstract

Amid the push for green transformation and industrial upgrades, how heavily polluting enterprises achieve green transition has become a focal point for the government and society. In response to policy demands, these enterprises often pursue green mergers as a means of transformation, though under varying conditions, they may opt for either green mergers or greenwashing, raising questions about their motivations. Based on evolutionary game theory, this paper constructs a four-party game model involving the media, government, public, and enterprises to analyze strategy stability. Using Lyapunov's first law, this paper examine the stability of equilibrium points, investigating how factors like media transparency, regulatory strength, and public discernment affect corporate decisions. The findings show that: 1) corporate green mergers are significantly influenced by media authenticity, government oversight, and public discernment; 2) media and public supervision play crucial roles in curbing greenwashing. Based on these insights, we propose strengthening media supervision, optimizing regulatory policies, and enhancing public environmental awareness to promote green mergers.

## 1. Introduction

Regional economic and social development has not only accelerated global industrialization and modernization but also placed immense pressure on environmental resources and ecosystems. The unexpected outbreak of the COVID-19 pandemic and frequent extreme climate events have further disrupted global economic and social stability, forcing nations to reassess their development models [1]. In response, many countries and regions have strengthened public health systems [2,3], promoted green energy and sustainable development strategies, and leveraged technological innovation to improve crisis response capabilities [4]. These measures have helped mitigate some of the adverse effects.

**Data availability statement:** All relevant data are within the manuscript and its Supporting Information files.

**Funding:** The author(s) received no specific funding for this work.

**Competing interests:** The authors have declared that no competing interests exist.

At the same time, the overlapping impacts of environmental issues have driven governments worldwide to prioritize the coordinated development of the economy, society, and environment as a critical future growth strategy. In China, the government has elevated ecological civilization to a national strategy. By improving environmental policy tools and implementing carbon neutrality goals, the government has made scientific and systematic environmental regulation a key means to achieving sustainable development [5,6].

Against this backdrop, the green transformation and upgrading of heavily polluting enterprises have become a focal point of policy and public attention. As environmental regulations tighten, these enterprises face increasing pressure to transform. However, they also receive policy incentives to reduce their environmental impact through green mergers and acquisitions (M&A), technological innovation, and other means [7–9]. Balancing regulatory constraints with incentives, while achieving green transformation, has emerged as a pressing issue for both academia and industry. Successful transformations not only contribute to ecological civilization but also provide valuable lessons and pathways for achieving harmonious economic and environmental development.

Green transformation is increasingly recognized as a key pathway for China's high-quality economic growth [10]. In the context of escalating global climate change and environmental crises, the sixth United Nations Environment Assembly emphasized the need for the international community to jointly address climate change, restore natural ecosystems, and strive for a pollution-free future. These goals have set a clear direction for global enterprises pursuing green development.

Green mergers and acquisitions, as a vital approach to driving green transformation, offer distinct advantages over traditional internal environmental investments and green innovation. Green M&A typically involves shorter cycles, lower risks, and faster integration of environmental resources and technological advantages [11,12]. By adopting a green development philosophy in target selection, deal structuring, and post-merger integration, green M&A enhances corporate energy efficiency and emissions reduction. It also sends a positive "carbon reduction" signal to the market, contributing to sustainable development goals [13]. For heavily polluting enterprises, green M&A represents a key pathway for achieving deeper environmental transformation and serves as a core policy tool for green resource allocation under China's ecological civilization and carbon neutrality objectives [14]. Consequently, green M&A has become a preferred strategy for advancing sustainable development, improving competitiveness, and supporting high-quality growth, making it a central focus of academic and practical research [15].

During the green transformation process, heavily polluting enterprises typically face two strategic options: On one hand, they can pursue green M&A to achieve resource integration, technological innovation, and brand enhancement, thereby securing long-term government subsidies and competitive market advantages. On the other hand, some may adopt short-term greenwashing strategies to evade regulation, lower costs, and reap short-term gains [16]. However, if greenwashing is truthfully reported by the media, it can lead to reputational risks, loss of market

share, and potentially severe government penalties. Clearly, this strategic decision is not merely a matter of internal cost-benefit analysis; it is shaped by external factors such as government regulation, media oversight, and public feedback, all of which interact dynamically.

Existing studies on corporate green transformation tend to take a single-dimensional perspective. Some research focuses on the incentives and penalties of government regulatory policies. For example, Cranston discussed the fundamental structure of regulatory systems and their application in environmental governance [17], while McKean examined enforcement costs and difficulties under stringent regulations [18]. These studies highlight the importance of external constraints imposed by government policy tools on corporate behavior. However, focusing solely on government perspectives fails to explain why enterprises continue to engage in greenwashing under real-world conditions. Corporate strategic choices depend not only on government policies but also on market information and social oversight. Other research examines the role of media in transmitting environmental information and monitoring enterprises. Bulger and Davison emphasized the media's positive role in enhancing corporate transparency and shaping public environmental awareness [19], while Caled and Silva explored the risks of misinformation in digital media and its impact on corporate behavior [20]. This body of literature demonstrates that media can expose corporate misconduct through truthful reporting. However, media may also collude with enterprises for financial gain, producing false reports and undermining external oversight. Additionally, some scholars have studied the public's role in environmental governance, highlighting how public environmental awareness [21] and information discernment can promote corporate environmental responsibility and exert external pressure on greenwashing behavior [22,23]. Nevertheless, research that focuses solely on public oversight often overlooks the interaction effects between public supervision, government regulation, and media reporting.

While these studies offer valuable insights, they often fail to treat government, enterprises, media, and the public as an integrated system. Consequently, they struggle to reveal the complex interaction mechanisms and strategy evolution processes among these stakeholders. In reality, corporate decision-making during green transformation results from the combined influence of government policies, media information dissemination, and public supervision. Only a comprehensive model that captures the dynamic feedback effects and adjustment processes among multiple stakeholders can provide a more accurate depiction of the green transformation landscape.

To address this theoretical gap, this study employs evolutionary game theory to develop a four-party game model involving government regulators, heavily polluting enterprises, media, and the public. The model assumes that all participants exhibit bounded rationality and approach optimal equilibrium solutions through iterative games under incomplete information conditions [24,25]. It incorporates key variables such as government regulation (strict versus lenient), corporate strategies (green M&A versus greenwashing), media coverage (truthful versus false reporting), and public oversight (information discernment and feedback). By doing so, the model dynamically depicts the interactions and strategic adjustments among these stakeholders.

In the four-party game model constructed in this study, the strategic choices of each stakeholder stem from an internal cost-benefit tradeoff during the green transformation process [26,27]. For heavily polluting enterprises, the decision often comes down to choosing between green mergers and acquisitions (M&A) and greenwashing strategies. Opting for green M&A allows these enterprises to integrate green resources, promote technological innovation, and potentially gain government subsidies and market recognition, thereby establishing a long-term competitive advantage [8]. In contrast, while greenwashing may yield short-term economic benefits, it carries significant risks of reputational damage, market sanctions, and legal consequences once exposed by the media [16]. Therefore, an enterprise's strategic decisions depend not only on internal cost-benefit considerations but also on external factors such as policies, regulatory pressures, and market feedback.

Secondly, the primary motivation of government regulators lies in improving environmental quality, promoting green industry development, and increasing revenues from environmental protection taxes and fines, thereby enhancing public

health and social welfare [17,18]. However, governments must also contend with high administrative costs and enforcement risks. As a result, they carefully balance strict and lenient regulation in order to maximize long-term environmental benefits and maintain social stability.

Media, as both an information disseminator and a key social watchdog, must balance its pursuit of commercial revenue with the need to maintain credibility. Truthful reporting on corporate misconduct can enhance the media's reputation and foster public trust [19], yet it may also lead to a loss of advertising and sponsorship income from those corporations. On the other hand, false reporting can generate short-term financial gains, but under strict government regulation and increased public scrutiny, it exposes the media to administrative penalties and reputational damage [20].

Finally, as the ultimate beneficiaries of environmental governance, the public is motivated to obtain accurate information and engage in oversight to ensure that companies fulfill their environmental responsibilities. Although this may involve certain time and effort costs, the public's objective evaluation of corporate environmental performance can help prevent greenwashing and encourage stricter government regulation [28]. Conversely, if the public blindly trusts the media or encounters opaque information channels, their supervision over polluting enterprises may weaken, potentially increasing environmental risks.

This study makes several contributions. First, it advances theoretical innovation. Prior research largely focused on individual stakeholders or bilateral relationships, such as the effects of government regulation on corporate behavior [17] or the role of media in information transmission [19]. This study, by constructing a four-party game model, integrates the interactions among government, enterprises, media, and the public into a unified framework. It reveals how these stakeholders continuously adjust their strategies over repeated games under conditions of limited information and rationality, ultimately achieving an overall equilibrium. This integrated approach provides new theoretical insights into the driving mechanisms behind green transformation [8,16]. Second, it introduces methodological innovation. By combining evolutionary game theory with replicator dynamics analysis, this study systematically describes the learning and adaptation processes of each stakeholder in a dynamic environment. Unlike traditional static models, this approach captures the strategic evolution trajectories that emerge as stakeholders receive feedback and update their information, thus offering a more accurate representation of the complexities involved in green transformation [19,20]. Additionally, this study employs both quantitative and qualitative methods, providing a theoretical and methodological foundation for subsequent empirical research. Lastly, it offers practical implications. Although this study uses China as a case for simulation analysis, the game model's fundamental logic—corporate cost-benefit tradeoffs, government regulation strategies, media oversight effects, and public feedback mechanisms—has universal applicability. For example, in regions like the European Union and North America, strict environmental regulations, carbon trading systems, and independent media supervision also play critical roles in promoting corporate green transformation. With minor adjustments to model parameters, this framework can guide green transformation efforts in other countries or regions, offering theoretical and policy recommendations [29].

In conclusion, this study's multi-stakeholder game model not only provides a theoretical explanation for the strategic choices involved in the green transformation of heavily polluting enterprises but also reveals the complex interactive mechanisms among external factors such as government regulation, media oversight, and public feedback. The findings contribute to the game theory literature on green transformation and environmental governance while offering practical policy insights for harmonizing diverse interests and achieving sustainable development across various contexts.

This paper is structured as follows: Section 2 outlines the problem and constructs the game model; Section 3 analyzes the stability of the stakeholders; Section 4 evaluates the stability of strategy combinations; Section 5 conducts simulations under different scenarios; Section 6 discusses the study's implications and future research directions; Section 7 summarizes the conclusions.

## 2. Problem description and modeling

### 2.1 Problem description

With the rapid development of green low-carbon economy and the rising public awareness of environmental protection, green M&A has become an important strategic choice for heavily polluting enterprises in market competition. Some heavily polluting enterprises actively promote green M&A, aiming to reduce pollution, enhance environmental benefits, and improve brand value and market competitiveness through practical environmental protection measures. However, there are also heavily polluting enterprises that choose to adopt the strategy of "greenwashing", attempting to mislead regulators and the public through superficial environmental protection initiatives or false publicity, so as to avoid the actual environmental responsibilities and costs. Based on this phenomenon, this study constructs a multi-party game model around the environmental behavior of heavy polluters, covering four major stakeholders, namely, the heavily polluting enterprise, the media, the government, and the public, and analyzes their complex interactions in the game. The behavioral choices and strategies of each party interact with each other and determine the final outcome of the game. This study focuses on the following three issues:

(1) How can a multi-party game mechanism reduce the occurrence of greenwashing behaviors by heavily polluting enterprises and encourage them to proactively choose green mergers and acquisitions?

(2) Under a combination of government incentives and constraints, how can the economic benefits of green mergers and acquisitions be enhanced to make them the priority choice for enterprises?

(3) How can the supervisory mechanisms of media and the public be leveraged to increase transparency in corporate environmental practices, effectively identifying and curbing greenwashing behaviors?

In this paper, we construct the logic diagram of multi-subject game as shown in Fig 1.

### 2.2 Model construction

**Hypothesis 1**: Participants in the Game. In this study, we selected government regulators, heavily polluting enterprises, media, and the public as the primary participants in the game model. This selection reflects the critical role these

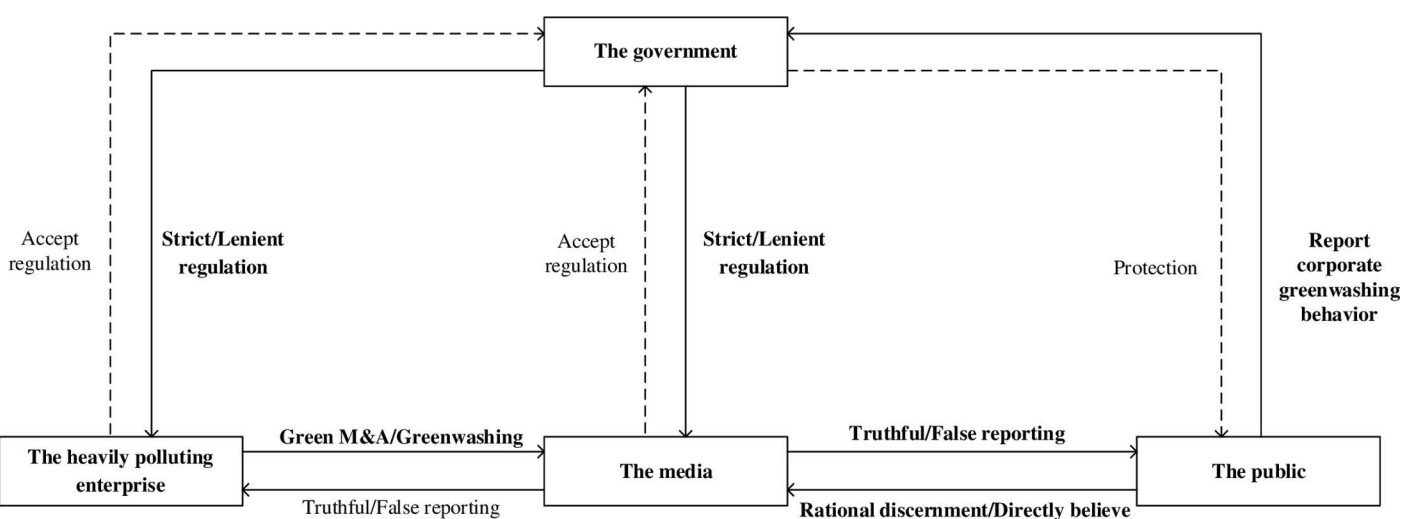

**Fig 1. Multi-subject game logic diagram.**

stakeholders play in environmental governance and green transformation. It is also strongly supported by evolutionary game theory and incomplete information game theory [24,25]. At the outset of the game, due to incomplete information and uncertainty about future economic trends, participants cannot accurately predict each other's behaviors. As a result, they must approach equilibrium solutions through multiple rounds of iteration, continuously adjusting their strategies based on feedback to maximize their own benefits. For heavily polluting enterprises, the choice between green M&A and green-washing revolves around the trade-off between long-term transformation benefits and short-term economic gains. Opting for green M&A can help integrate green resources and enhance technological innovation capabilities, while also securing government policy support and subsidies [16]. Conversely, greenwashing may temporarily reduce costs and boost profits, but once exposed by truthful media reports, reputation risks and market share losses become substantial. Under conditions of bounded rationality, enterprises learn and adjust their strategies through iterative gameplay to maximize their overall benefits. The government's primary goals include improving environmental quality, enhancing social fairness, and increasing public satisfaction, while also balancing regulatory costs and political risks. Strict regulation can yield environmental improvements and public support [17], but it also entails significant enforcement and administrative costs [18]. Consequently, the government must weigh strict and lenient regulation, adjusting its strategies throughout the iterative game process to achieve long-term governance goals. Media, as a critical channel for information dissemination and oversight, must balance economic returns, reputational effects, and external regulatory pressures. Truthful reporting enhances media credibility and long-term brand value [19], but may forfeit short-term revenue from corporate partnerships. False reporting, on the other hand, may provide direct economic benefits (such as corporate bribes), but it risks administrative penalties and reputational damage under strict government regulation and heightened public information discernment [20]. Media must adapt their reporting strategies based on external feedback to achieve a balance between economic and social returns in the iterative game. The public contributes to the game through information discernment and supervisory feedback. To accurately judge the authenticity of media reports, the public incurs certain time and effort costs. When the public detects corporate greenwashing, they may report these practices to the government, prompting further regulatory actions [19]. In turn, public oversight pressures corporations and media to correct deviations, creating positive incentives throughout the iterative game process. With bounded rationality, the public gradually refines their information judgment skills, influencing corporate strategies and shaping government and media regulatory approaches.

Hypothesis 2: Strategy combination of the game. This paper focuses on the strategic choices made by the other three parties of the game when the heavy polluting enterprise chooses green mergers and acquisitions. So the strategy combinations in this paper are the heavily polluting enterprise (green mergers and acquisitions, greenwashing), the media (true reporting, false reporting), the government (strict regulation, loose regulation), and the public (objective identification, direct belief).

**Hypothesis 3**: The choice proportion of game strategies. The probability that a heavily polluting enterprise chooses green M&A is $e$ $(0 \leq e \leq 1)$, and the probability that it chooses greenwashing is $(1-e)$; the probability that the media chooses truthful reporting is $m (0 \leq m \leq 1)$, and the probability that it chooses false reporting is $(1-m)$; the probability that the government chooses strict regulation is $g (0 \leq g \leq 1)$, and the probability that it chooses lenient regulation is $(1-g)$; the probability that the public chooses objective identification is $p (0 \leq p \leq 1)$, and the probability that it chooses to believe it directly is $(1-p)$.

**Hypothesis 4**: For heavily polluting enterprises, the choice between green mergers and acquisitions (M&A) and green-washing strategies essentially involves weighing long-term strategic benefits against short-term gains. Opting for green M&A yields immediate benefits $Q_s$ [30,31], but also incurs merger costs $Q_h$ [30]. Green M&A helps enterprises rapidly integrate green resources, enhance green innovation capabilities, and comply with strict government environmental regulations [8]. Additionally, policies such as carbon trading systems provide further support through subsidies or tax incentives, increasing the economic returns of green M&A [32]. These forms of government support are captured in the model as a policy subsidy parameter $Q_c$ [33]. In contrast, while greenwashing can deliver short-term gains $Q_f$, it involves certain costs

$Q_k$ [34], including internal governance expenses and potential legal risks. If the media truthfully report the greenwashing, the enterprise faces serious reputational damage and market penalties $Q_d$ [16]. To avoid these risks, some enterprises may bribe the media (bribe amount parameter $M_s$ [35]) in an attempt to suppress the truth and temporarily evade scrutiny from the public and regulators [20,36]. Moreover, under strict government regulation, enterprises found engaging in greenwashing are subject to administrative penalties (penalty amount parameter $L_s$), further reducing their net profits. Thus, in making their strategic decisions, heavily polluting enterprises must balance the long-term technological and branding benefits of green M&A—along with positive policy incentives—against the reputational and market risks associated with greenwashing. This interplay of multiple external factors drives enterprises to rationally choose between green M&A and greenwashing strategies, ultimately aiming to maximize overall returns.

**Hypothesis 5**: In reporting corporate activities, media obtain a base revenue, denoted as $M_w$ which includes income from advertisements, subscription fees, and other business collaborations. When media choose truthful reporting, their objective and impartial reporting style can earn widespread social and public recognition, significantly enhancing their reputation value, denoted as $M_c$ [37]. This reputational gain helps build long-term brand trust and sustained collaborative relationships. However, exposing corporate greenwashing through truthful reporting may lead to reduced corporate partnerships or lower sponsorship amounts, causing a direct revenue loss, denoted as $Z_s$. On the other hand, if media opt for false reporting, they may receive additional short-term financial gains from corporate bribes, denoted as $M_s$ [35]. However, under strict government regulation, such behavior can result in administrative penalties, denoted as $L_k$ [38]. Moreover, when the public demonstrates strong information discernment and rationally identifies the media's false reporting, the media can suffer further reputational damage, denoted as $L_c$ [20]. This negative impact not only weakens the media's long-term competitiveness but may also prompt more stringent regulatory measures, further restricting their market share [36].

**Hypothesis 6**: When the government adopts a strict regulatory strategy, it receives basic revenue $A_c$ [39,40], while also gaining added benefits $R_c$ [37]from reduced environmental pollution, improved public health, and enhanced ecosystem quality. These positive outcomes not only encourage corporate green transformation but also generate long-term social feedback, bolstering government credibility and policy support [17]. However, strict regulation incurs substantial enforcement costs $A_s$ [41] associated with human resources, material inputs, and time. These costs represent a significant burden for the government. Faced with these ongoing expenses, the government may be inclined to choose a more lenient regulatory strategy, which can lower administrative expenditures and stimulate corporate economic activity in the short term. These short-term gains are denoted as $I_p$ [42]. Nonetheless, lenient regulation often leads to environmental degradation and negative public opinion regarding ineffective government oversight. This results in additional social and political losses, represented as $I_n$ [17].As a result, the government must find a balance between the long-term environmental benefits of strict regulation and the short-term economic advantages of leniency. Achieving this balance involves weighing the direct costs and added benefits of each policy approach, while also considering the external social and market feedback's impact on government credibility, thereby ensuring sustainable public governance.

**Hypothesis 7**: For the public, the positive effects of corporate green M&A include improved environmental quality, enhanced public health, and increased social welfare, collectively denoted as $G$ [43]. However, according to information asymmetry theory, the public must expend time and effort to verify the authenticity and completeness of media reports, incurring an identification cost $H_c$ [19].If the public objectively identifies corporate greenwashing, they may, with a certain probability $\beta$ [44], file complaints with the government and receive compensation $T_g$ [45–47] from the enterprises involved, partially offsetting the negative impacts of environmental degradation. Conversely, as the relatively disadvantaged party in terms of information access, if the public blindly trusts the media's false reports without conducting objective verification, they may suffer economic and environmental losses, denoted as $H_s$ [48], due to misleading information. This cost-benefit trade-off shows that while public supervision can encourage enterprises to adopt green transformation measures, it also entails significant costs and risks associated with information discernment.

Construct a parameter definition table(Table 1) based on the above parameter definitions.

**2.3 Payoff matrix Table 2**

### 3. Analysis of the stability of each game subject's strategy

#### 3.1 Stability analysis of the merger and acquisition strategy for the heavily polluting enterprise

Based on the asymmetric replicator dynamic evolution method and Table 2, the expected returns for the Heavily Polluting Enterprise choosing green M&A or greenwashing strategies are $U_e$ and $U_{1-e}$, respectively. The replicator dynamic equation and the first-order derivative of their strategic behaviors are shown in Equations (2) and (3).

$$\begin{cases} U_e = Q_s - Q_h + gQ_c \\ U_{1-e} = Q_f - M_s - Q_k + m(M_s - Q_d) - g(L_s + p\beta T_g) \end{cases} \tag{1}$$

$$\begin{aligned} F(e) &= \frac{de}{dt} = e(U_e - \bar{U}) = e(1-e)(U_e - U_{1-e}) \\ &= e(1-e)[M_s - Q_f - Q_h + Q_s + Q_k + m(Q_d - M_s) + g(L_s + Q_c + p\beta T_g)] \end{aligned} \tag{2}$$

$$F'(e) = (1-2e)[M_s - Q_f - Q_h + Q_s + Q_k + m(Q_d - M_s) + g(L_s + Q_c + p\beta T_g)] \tag{3}$$

According to the stability theorem of differential equations, the probability that the heavily polluting enterprise chooses to make a green merger is in a steady state needs to satisfy: $F(e) = 0$ and $F'(e) < 0$.

Proposition 1: When $m > m_0$, the heavily polluting enterprise's stabilizing strategy is to engage in green mergers and acquisitions; when $m < m_0$, the heavily polluting enterprise's stabilizing strategy is to engage in greenwashing; when $m = m_0$, the stabilizing strategy cannot be determined.

Where the threshold $m_0 = \frac{M_s - Q_f - Q_h + Q_k + Q_s + g(L_s + Q_c + p\beta T_g)}{M_s - Q_d}$.

Proof: Let $N(m) = M_s - Q_f - Q_h + Q_s + Q_k + m(Q_d - M_s) + g(L_s + Q_c + p\beta T_g)$, $\partial N(m)/\partial(m) > 0$, so $N(m)$ is an increasing function on $m$, when $m > m_0$ and $N(m) > 0$, $F(e)\big|_{e=1} = 0$ and $F'(e)\big|_{e=1} < 0$, then $e = 1$ has stability; when $m < m_0$ and $N(m) < 0$, $F(e)\big|_{e=0} = 0$ and $F'(e)\big|_{e=0} < 0$, then $e = 0$ has stability. When $m = m_0$, $N(m) = 0$, $F(e) = 0$ and $F'(e) = 0$, at this time the heavily polluting enterprise can not determine the stabilization strategy.

Proposition 1 suggests that when the probability of media's true reporting on the strategic choices of enterprises rises, the heavily polluting enterprise will choose green M&A strategy more actively. Of course, when the media's strategic choices of enterprises are false reports, the strategic choices of the heavily polluting enterprise will gradually tend to be greenwashed. Therefore, the media's false reporting behavior is not conducive to the heavily polluting enterprise's green M&A, and the government should increase the supervision and pay attention to the media's influence effect, which is of great significance in promoting the heavily polluting enterprise's green M&A.

The phase diagram of the heavily polluting enterprise's strategy choices according to Proposition 1 is shown below:

From the Fig 2, the $V_{e0}$ part of the volume is the probability that the heavily polluting enterprise chooses greenwashing, and the $V_{e1}$ part is the probability that the heavily polluting enterprise chooses green mergers and acquisitions, which can be calculated as follows:

$$\begin{aligned} V_{e0} &= \int_0^1 \int_0^1 \frac{M_s - Q_f - Q_h + Q_k + Q_s + g(L_s + Q_c + p\beta T_g)}{M_s - Q_d} \, dg \, de \\ &= \frac{1/2(L_s + Q_c) + M_s - Q_f - Q_h + Q_k + Q_s + 1/2p\beta T_g}{M_s - Q_d} \end{aligned} \tag{4}$$

**Table 1. Parameter definition table.**

| Parameters | Definition | Ranges |
|---|---|---|
| $e$ | the probability that the heavily polluting enterprise chooses green M&A | $0 \leq e \leq 1$ |
| $m$ | the probability that the media chooses truthful reporting | $0 \leq m \leq 1$ |
| $g$ | the probability that the government chooses strict regulation | $0 \leq g \leq 1$ |
| $p$ | the probability that the public chooses objective identification | $0 \leq p \leq 1$ |
| $Q_s$ | the direct benefits that heavily polluting enterprises gain from choosing green M&A | $(0, +\infty)$ |
| $Q_h$ | the costs that heavily polluting enterprises incur when choosing green M&A | $(0, +\infty)$ |
| $Q_c$ | government subsidies for the heavily polluting enterprise | $(0, +\infty)$ |
| $Q_f$ | the benefits that the heavily polluting enterprise gain from choosing greenwashing | $(0, +\infty)$ |
| $Q_k$ | the direct costs that the heavily polluting enterprise incur when choosing greenwashing | $(0, +\infty)$ |
| $Q_d$ | the losses incurred by the heavily polluting enterprise due to truthful media reporting of their greenwashing practices | $(0, +\infty)$ |
| $M_s$ | the amount of bribery given by the heavily polluting enterprise to the media | $(0, +\infty)$ |
| $L_s$ | the penalty amount imposed by the government on the heavily polluting enterprise | $(0, +\infty)$ |
| $M_w$ | the basic revenue gained by the media from reporting on corporate actions | $(0, +\infty)$ |
| $M_c$ | the reputation gain acquired by the media | $(0, +\infty)$ |
| $Z_s$ | the loss of cooperative revenue for the media from partnerships with heavily polluting enterprises | $(0, +\infty)$ |
| $L_k$ | the amount of government penalties faced by the media for false reporting | $(0, +\infty)$ |
| $L_c$ | the reputational loss faced by the media | $(0, +\infty)$ |
| $A_c$ | the basic revenue gained by the government from strict regulation | $(0, +\infty)$ |
| $R_c$ | the additional revenue gained by the government from strict regulation | $(0, +\infty)$ |
| $A_s$ | the cost of strict regulation for the government | $(0, +\infty)$ |
| $I_p$ | the basic revenue gained by the government from lenient regulation | $(0, +\infty)$ |
| $I_n$ | the losses faced by the government from lenient regulation | $(0, +\infty)$ |
| $G$ | the positive effect of a enterprise's green M&A on the public | $(0, +\infty)$ |
| $H_c$ | the public's discernment cost | $(0, +\infty)$ |
| $\beta$ | the probability of the public filing complaints to the government | $[0, 1]$ |
| $T_g$ | the compensation amount given by the heavily polluting enterprise to the public | $(0, +\infty)$ |
| $H_s$ | the loss incurred by the public due to misinformation | $(0, +\infty)$ |

$$V_{e1} = 1 - V_{e0}$$

$$= \frac{Q_f + Q_h - Q_k - Q_s - Q_d - 1/2(L_s + Q_c - p\beta T_g)}{M_s - Q_d} \tag{5}$$

Corollary 1.1: When the public actively chooses to rationally identify the truthfulness of media reports, enterpsies will be more cautious in choosing green M&A.

Proof: The probability that the heavily polluting enterprise choose green M&A is $V_{e1}$, and the partial derivation of $p$ from $V_{e1}$ yields $\partial V_{e1}/\partial p > 0$, so as the probability that the public chooses a rational identification strategy rises, the heavily polluting enterprise will be more inclined to choose the green M&A strategy.

Corollary 1.2: When the probability of the public reporting the heavily polluting enterprise to the government for greenwashing behavior increases, enterprises respond. This increase is due to the public's rational discernment of media reports and the discovery that enterprises are indeed engaging in greenwashing. As a result, enterprises will shift to choosing green M&A strategies.

**Table 2. The payment matrix of the game among the heavily polluting enterprise, the media, the government and the public.**

| Strategy Selection | | Media | Government | | | |
|---|---|---|---|---|---|---|
| | | | **Strict regulation** $g$ | | **Lenient regulation** $1-g$ | |
| | | | **Public: Rational discernment** $p$ | **Public: Directly believe** $1-p$ | **Public: Rational discernment** $f$ | **Public: Directly believe** $1-f$ |
| The heavily polluting enterprise | GreenM&A $e$ | Truthful reporting $m$ | $Q_s + Q_c - Q_h$ $M_w + M_c$ $A_c + R_c - A_s$ $G - H_c$ | $Q_s + Q_c - Q_h$ $M_w + M_c$ $A_c + R_c - A_s$ $G$ | $Q_s - Q_h$ $M_w + M_c$ $I_p - I_n$ $G - H_c$ | $Q_s - Q_h$ $M_w + M_c$ $I_p - I_n$ $G$ |
| | | False reporting $1-m$ | $Q_s + Q_c - Q_h$ $M_w - L_c - L_k$ $A_c + R_c - A_s + L_k$ $G - H_c$ | $Q_s + Q_c - Q_h$ $M_w - L_k$ $A_c + R_c - A_s + L_k$ $G - H_s$ | $Q_s - Q_h$ $M_w - L_c$ $I_p - I_n$ $G - H_c$ | $Q_s - Q_h$ $M_w$ $I_p - I_n$ $G - H_s$ |
| | Green washing $1-e$ | Truthful reporting $m$ | $Q_f - Q_k - Q_d - L_s - \beta T_g$ $M_w + M_c - Z_s$ $A_c + R_c + L_s - A_s$ $\beta T_g - H_c$ | $Q_f - Q_k - Q_d - L_s$ $M_w + M_c - Z_s$ $A_c + R_c + L_s - A_s$ $0$ | $Q_f - Q_k - Q_d$ $M_w + M_c - Z_s$ $I_p - I_n$ $-H_c$ | $Q_f - Q_k - Q_d$ $M_w + M_c - Z_s$ $I_p - I_n$ $0$ |
| | | False reporting $1-m$ | $Q_f - Q_k - M_s - L_s - \beta T_g$ $M_w - L_c - L_k + M_s$ $A_c + R_c - A_s + L_s + L_k$ $\beta T_g - H_c$ | $Q_f - Q_k - M_s - L_s$ $M_w - L_k + M_s$ $A_c + R_c - A_s + L_s + L_k$ $-H_s$ | $Q_f - Q_k - M_s$ $M_w - L_c + M_s$ $I_p - I_n$ $-H_c$ | $Q_f - Q_k - M_s$ $M_w + M_s$ $I_p - I_n$ $-H_s$ |

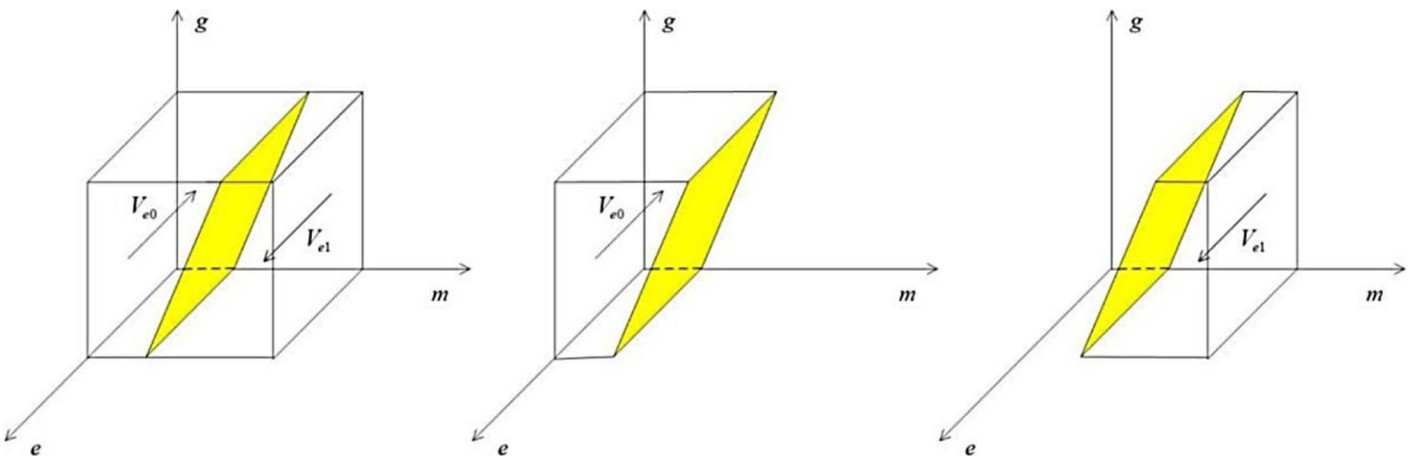

**Fig 2. Phase diagram of strategy choices for the heavily polluting enterprise.**

Proof: The probability that the heavily polluting enterprise chooses a green M&A is $V_{e1}$, and the partial derivation of $\beta$ from $V_{e1}$ yields $\partial V_{e1}/\partial \beta > 0$, so as the probability of public reporting rises, enterprises will be more inclined to choose a green M&A strategy.

## 3.2 Stability analysis of media coverage strategies

Based on the asymmetric replicator dynamic evolution method, the expected returns for the media choosing truthful reporting or false reporting strategies are $U_m$ and $U_{1-m}$, respectively. The replicator dynamic equation and the first-order derivative of their strategic behaviors are shown in Equations (7) and (8).

$$\begin{cases} U_m = M_c + M_w + (e-1)Z_s \\ U_{1-m} = M_s + M_w - pL_c - eM_s - gL_k \end{cases}$$

(6)

$$F(m) \quad = \frac{dm}{dt} = m(U_m - \bar{U}) = m(1-m)(U_m - U_{1-m})$$

$$= m(1-m)[M_c - M_s - Z_s + pL_c + e(Z_s + M_s) + gL_k]$$

(7)

$$F'(m) = (1-2m)[M_c - M_s - Z_s + pL_c + e(Z_s + M_s) + gL_k]$$

(8)

According to the stability theorem of differential equations, the probability that the media chooses to carry out truthful reporting is in a steady state needs to be satisfied: $F(m) = 0$ and $F'(m) = 0$.

Proposition 2:The stabilizing strategy of the media is to choose truthful reporting when $e > e_0$, the stabilizing strategy of the media is to engage in false reporting when $e < e_0$, and the stabilizing strategy cannot be determined when $e = e_0$.

Where the threshold $e_0 = \frac{-M_c + M_s + Z_s - pL_c - gL_k}{M_s + Z_s}$.

Proof: Let $N(e) = M_c - M_s - Z_s + pL_c + e(Z_s + M_s) + gL_k$, so $N(e)$ is an increasing function on $e$, when $e > e_0$, $N(e) > 0$, $F(m)|_{m=1} = 0$ and $F'(m)|_{m=1} < 0$, then $m = 1$ has stability; when $e < e_0$, $N(e) < 0$, $F(m)|_{m=0} = 0$ and $F'(m)|_{m=0} < 0$, then $m = 0$ has stability. When $e = e_0$, $N(e) = 0$, $F(m) = 0$ and $F'(m) = 0$, at this time the media can not determine the stabilization strategy.

Proposition 2 suggests that a rise in the probability of the heavily polluting enterprise choosing a green M&A strategy does lead to a positive choice of truthful reporting by the media; similarly a fall in the probability of the heavily polluting enterprise choosing a green M&A strategy leads to a gradual shift in the media's strategic choices in favor of false reporting.

Construct a media strategy selection phase diagram based on Proposition 2 as follows:

From the Fig 3, we can see that the $V_{m0}$ part of the volume is the probability of false reporting by the media, and the $V_{m1}$ part of it is the probability of true reporting by the media, which can be calculated as follows:

$$V_{m0} \quad = \int_0^1 \int_0^1 \frac{-M_c + M_s + Z_s - pL_c - gL_k}{M_s + Z_s} dgdm$$

$$= 1 - \frac{1/2L_k + M_c + pL_c}{M_s + Z_s}$$

(9)

$$V_{m1} \quad = 1 - V_{m0}$$

$$= \frac{1/2L_k + M_c + pL_c}{M_s + Z_s}$$

(10)

Inference 2.1: When the public chooses rational discernment toward reports on the heavily polluting enterprise, the likelihood of the media choosing truthful reporting increases. However, when the public adopts a stance of blind belief in such reports, the media is more likely to shift toward a false reporting strategy.

Proof: The $V_{m1}$ part is known to be the probability of truthful reporting by the media, and solving for the partial derivative of $V_{m1}$ with respect to $p$ yields $\partial V_{m1}/\partial p > 0$. Therefore, as the probability of the public choosing to rationally analyze the truthfulness of a report increases, the media will be more inclined to choose truthful reporting.

Inference 2.2: The media will choose the truthful reporting strategy more aggressively when the loss of trust faced by the media due to false reporting increases, when the reputational gains obtained by the media from truthful reporting increase, and when the government's penalties for false reporting in the media rise.

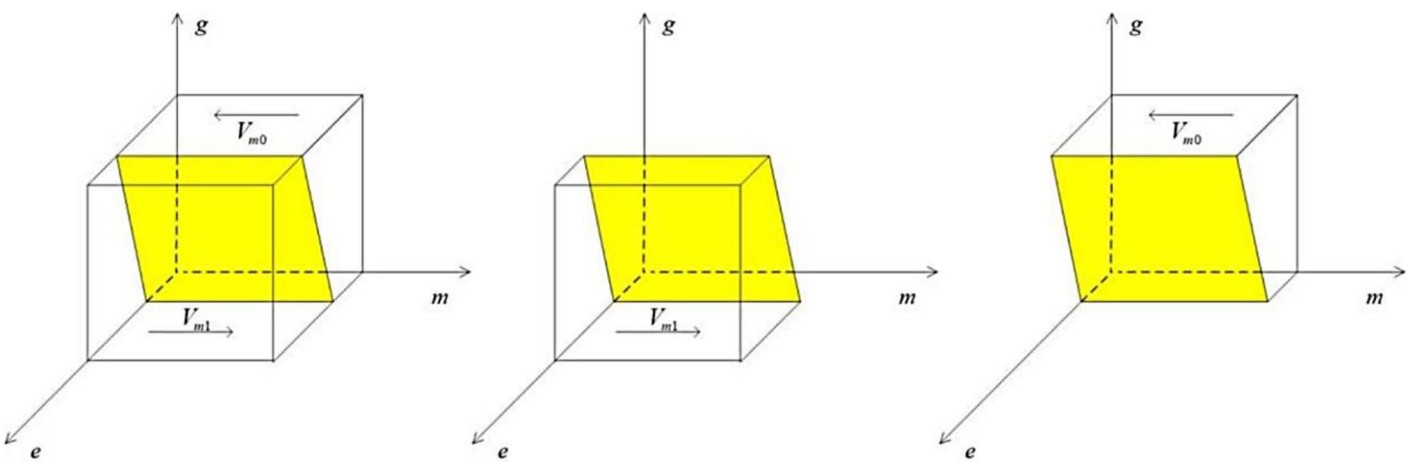

**Fig 3. Phase diagram of media strategy choices.**

Proof: the known part $V_{m1}$ is the probability of a true media report, and solving for the partial derivative $V_{m1}$ about $L_c$, $M_c$, $L_k$, respectively, yields $\partial V_{m1}/\partial L_c > 0$, $\partial V_{m1}/\partial M_c > 0$, $\partial V_{m1}/\partial L_k > 0$, which $V_{m1}$ is therefore positively correlated with $L_c$, $M_c$, $L_k$, respectively.

### 3.3 Stability analysis of government regulatory strategies

Based on the asymmetric replicator dynamic evolution method, the expected returns for the government choosing strict regulation or lenient regulation strategies are $U_g$ and $U_{1-g}$, respectively. The replicator dynamic equation and the first-order derivative of their strategic behaviors are shown in Equations (12) and (13).

$$\begin{cases} U_g = A_c - A_s + L_k + L_s + R_c - mL_k - eL_s \\ U_{1-g} = I_p - I_n \end{cases} \tag{11}$$

$$F(g) \quad = \frac{dg}{dt} = g(U_g - \bar{U}) = g(1-g)(U_g - U_{1-g})$$

$$= g(1-g)(A_c - A_s + L_k + L_s + R_c - mL_k - eL_s + I_n - I_p) \tag{12}$$

$$F'(g) = (1-2g)(A_c - A_s + L_k + L_s + R_c - mL_k - eL_s + I_n - I_p) \tag{13}$$

Proposition 3: When $e < e_1$, the government's stabilization strategy was to engage in strict regulation; when $e > e_1$, the government's stabilization strategy was to engage in lenient regulation; and when $e = e_1$, it was not possible to determine the stabilization strategy.

Where the threshold is $e_1 = \frac{A_c - A_s + I_n - I_p + L_k + L_s + R_c - mL_k}{L_s}$.

Proof: Let $M(e) = A_c - A_s + L_k + L_s + R_c - mL_k - eL_s + I_n - I_p$, $\partial M(e)/\partial(e) < 0$, therefore, $M(e)$ is a reduced function about $e$, when $e < e_1$, $M(e) > 0$, $F(g)\big|_{g=1} = 0$ and $F'(g)\big|_{g=1} < 0$, then $g = 1$ has stability. When $e > e_1$, $M(e) < 0$, $F(g)\big|_{g=0} = 0$ and $F'(g)\big|_{g=0} < 0$, then $g = 0$ has stability. When $e = e_1$, $M(e) = 0$, $F(g) = 0$ and $F'(g) = 0$, at that time, the government cannot determine the stabilization strategy.

Construct a government strategy phase diagram based on Proposition 3 as follows:

From the Fig 4, the $V_{g0}$ part of the volume is the probability that the government regulates loosely, and the $V_{g1}$ part is the probability that the government regulates strictly, which can be computed as:

$$V_{g1} = \int_0^1 \int_0^1 \frac{A_c - A_s + I_n - I_p + L_k + L_s + R_c - mL_k}{L_s} \, dm \, dg$$

$$= \frac{A_c - A_s + I_n - I_p + 1/2L_k + R_c}{L_s} + 1 \tag{14}$$

$$V_{g0} = 1 - V_{g1}$$

$$= -\frac{A_c - A_s + I_n - I_p + 1/2L_k + R_c}{L_s} \tag{15}$$

Inference 3.1: When the net benefits of strict regulation increase, the negative losses from lenient regulation rise, the short-term gains from lenient regulation decrease, penalties for media false reporting intensify, and social support for proactive regulation grows, the government will be more inclined to choose a proactive regulatory strategy.

Proof: Knowing that $V_{g1}$ is partially the probability of strict government regulation, take the partial derivatives of $V_{g1}$ with respect to $A_c - A_s, I_n, I_p, L_k, R_c$ to obtain $\partial V_{g1}/\partial(A_c - A_s) > 0, \partial V_{g1}/\partial I_n > 0, \partial V_{g1}/\partial I_p < 0, \partial V_{g1}/\partial L_k > 0, \partial V_{g1}/\partial R_c > 0$.

### 3.4 Public strategy stabilization analysis

Based on the asymmetric replicator dynamic evolution method, the expected returns for the public choosing rational discernment or blind belief strategies are $U_p$ and $U_{1-p}$, respectively. The replicator dynamic equation and the first-order derivative of their strategic behaviors are shown in Equations (17) and (18).

$$\begin{cases} U_p = eG - H_c + g(1-e)\beta T_g \\ U_{1-p} = (m-1)H_s + eG \end{cases} \tag{16}$$

$$F(p) = \frac{dp}{dt} = p(U_p - \bar{U}) = p(1-p)(U_p - U_{1-p})$$

$$= p(1-p)[-H_c + (1-m)H_s + g(1-e)\beta T_g] \tag{17}$$

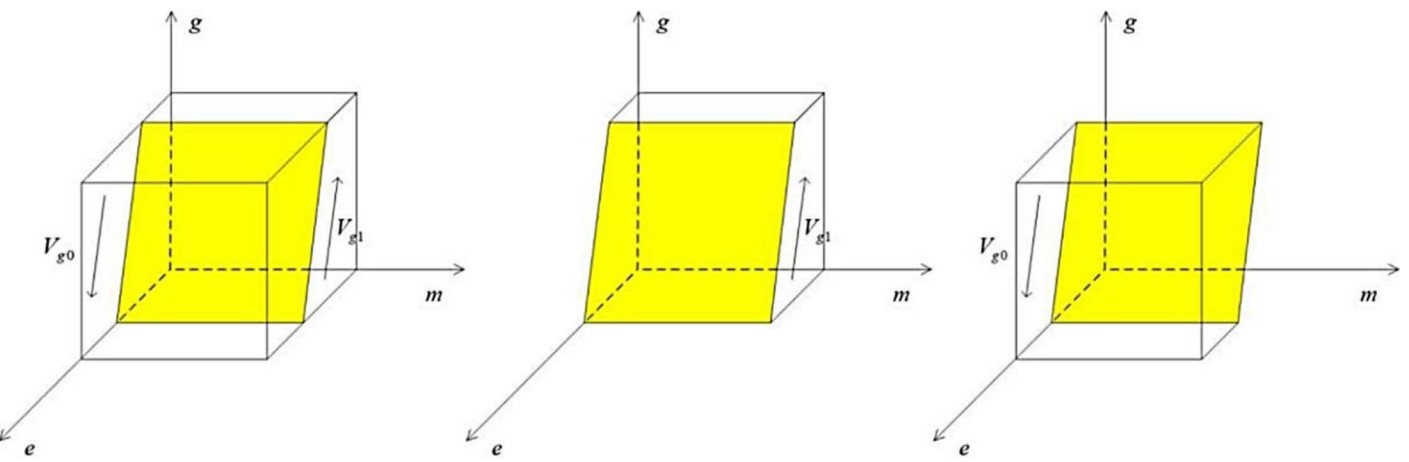

**Fig 4. The phase diagram of the government's strategy choices.**

$$F'(p) = (1-2p)[-H_c + (1-m)H_s + g(1-e)\beta T_g]$$
(18)

Proposition 4: When $m < m_1$, the public's stabilizing strategy was to engage in objective identification; when $m > m_1$, the public's stabilizing strategy was to engage in direct belief; and when $m = m_1$, it was not possible to determine the stabilizing strategy.

Where the threshold $m_1 = \frac{H_s - H_c + (1-e)g\beta T_g}{H_s}$.

Proof: Let $M(m) = -H_c + (1-m)H_s + g(1-e)\beta T_g$, $\partial M(m)/\partial m < 0$, $M(m)$ is a reduced function about $m$, when $m < m_1$, $M(m) > 0$, $F(p)\big|_{p=1} = 0$ and $F'(p)\big|_{p=1} < 0$, then there is stability for $p = 1$; when $m > m_1$, $M(m) < 0$, $F(p)\big|_{p=0} = 0$ and $F'(p)\big|_{p=0} < 0$, then there is stability on $p = 0$. When $m = m_1$, $M(m) = 0$, $F(p) = 0$ and $F'(p)\big|_{p=0} < 0$, at this time the public can not determine the stabilization strategy. Prove that.

The phase diagram of the public's strategy choice according to Proposition 4 is shown below:

As can be seen from the Fig 5, the $V_{p0}$ part of the volume is the probability that the public chooses to believe directly, and the $V_{p1}$ part of it is the probability that the public chooses to recognize it objectively, which can be calculated as:

$$V_{p1} = \int_0^1 \int_0^1 \frac{H_s - H_c + (1-e)g\beta T_g}{H_s} \, de \, dp$$

$$= 1 - \frac{H_c - 1/2g\beta T_g}{H_s}$$
(19)

$$V_{p0} = 1 - V_{p1}$$

$$= \frac{H_c - 1/2g\beta T_g}{H_s}$$
(20)

Corollary 4.1: The probability of the public choosing objective identification increases as the cost to the public of objectively determining the truthfulness of media reports decreases and the amount of compensation received rises.

Proof: $V_{p1}$ is known to be partly the probability that the government carries out strict regulation, and taking partial derivatives of $V_{p1}$ with respect to $H_c$ and $T_g$ respectively, we get $\partial V_{p1}/\partial H_c < 0$, $\partial V_{p1}/\partial T_g > 0$.

Corollary 4.2 When the public's losses due to direct belief in media reports rise, the public chooses a more cautious objective identification strategy.

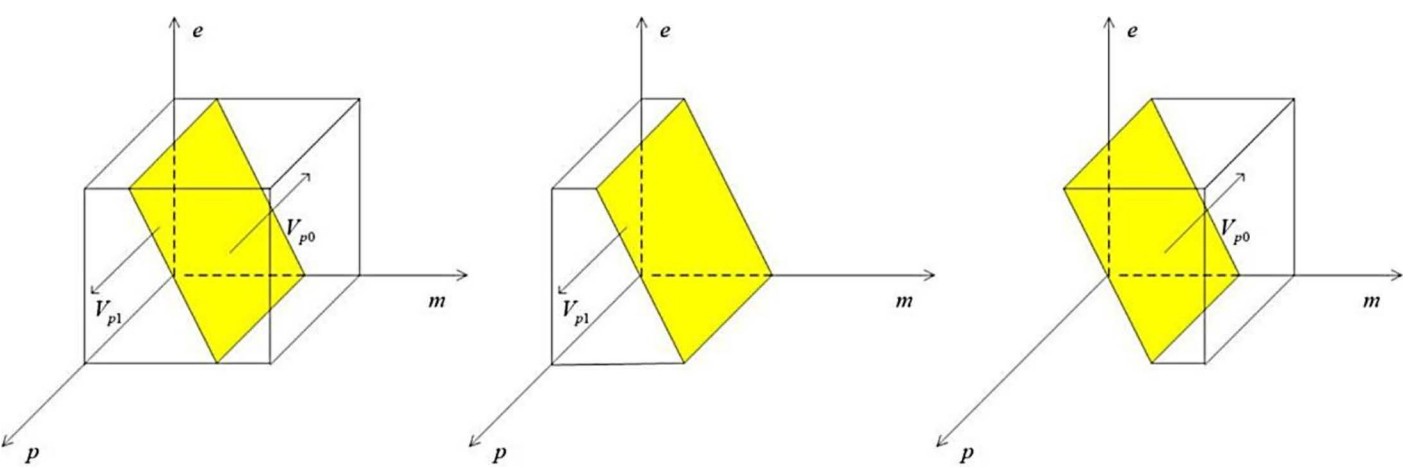

**Fig 5. The phase diagram of the public's strategy choices.**

Proof: the $V_{p1}$ part is known to be the probability that the government carries out strict regulation, and the partial derivatives of $V_{p1}$ with respect to $H_s$, respectively, give $\partial V_{p1}/\partial H_s > 0$.

Corollary 4.3 When government regulation increases, the public will also tend to choose the strategy of rationally discerning the truthfulness of media reports.

Proof: knowing that $V_{p1}$ is partially the probability that the government engages in strict regulation, take a partial derivative of $V_{p1}$ with respect to $g$ to get $\partial V_{p1}/\partial g > 0$.

## 4. Strategy portfolio stability point analysis

In the replicated dynamic system composed of four parties: enterprise, media, government and public, the stability of the strategy combinations of the four parties can be judged according to Lyapunov's first law. Therefore, in the replication dynamic system, this paper analyzes the stability of 16 pure strategy equilibrium points.

Based on the replication dynamic equations for each subject, the Jacobian matrix of the replication dynamic system is obtained as:

$$J = \begin{bmatrix} \partial F(e)/\partial e & \partial F(e)/\partial m & \partial F(e)/\partial g & \partial F(e)/\partial p \\ \partial F(m)/\partial e & \partial F(m)/\partial m & \partial F(m)/\partial g & \partial F(m)/\partial p \\ \partial F(g)/\partial e & \partial F(g)/\partial m & \partial F(g)/\partial g & \partial F(g)/\partial p \\ \partial F(p)/\partial e & \partial F(p)/\partial m & \partial F(p)/\partial g & \partial F(p)/\partial p \end{bmatrix}$$

Solve for the 16 eigenvalues respectively: $E_1(0,0,0,0)$, $E_2(0,1,0,0)$, $E_3(0,0,1,0)$, $E_4(0,1,1,0)$, $E_5(1,0,0,0)$, $E_6(1,1,0,0)$, $E_7(1,0,1,0)$, $E_8(1,1,1,0)$, $E_9(0,0,0,1)$, $E_{10}(0,1,0,1)$, $E_{11}(0,0,1,1)$, $E_{12}(0,1,1,1)$, $E_{13}(1,0,0,1)$, $E_{14}(1,1,0,1)$, $E_{15}(1,0,1,1)$, $E_{16}(1,1,1,1)$.

According to Lyapunov's First Method, if the real parts of all eigenvalues of the Jacobian matrix are negative, the equilibrium point is locally asymptotically stable; that is, the system will asymptotically converge to this equilibrium point when it approaches the vicinity of the point. If there exists an eigenvalue with a positive real part, the equilibrium point is unstable, meaning the system will diverge from the equilibrium rather than converge to it. If all eigenvalues have non-positive real parts, with at least one equal to zero, the stability of the equilibrium point is indeterminate, requiring further analysis.

Based on Table 3., the 16 pure strategy equilibria described above are analyzed, the only $E_1(0,0,0,0)$, $E_2(0,1,0,0)$, $E_3(0,0,1,0)$, $E_4(0,1,1,0)$, $E_6(1,1,0,0)$, $E_8(1,1,1,0)$, $E_9(0,0,0,1)$, $E_{11}(0,0,1,1)$, $E_{12}(0,1,1,1)$ are stabilized when the following conditions are met.

In Condition ①, an enterprise chooses the greenwashing strategy when it satisfies $Q_s + Q_k < Q_f + Q_h - M_s$, when the net benefit to the enterprise from making a green merger is less than the net benefit from making a greenwashing; When the media satisfy $M_c - Z_s < M_s$, i.e., when the additional benefit to the media from truthful reporting is less than the amount of corporate bribe received, the media will choose to report falsely; when the government satisfies $A_c + R_c - A_s + L_s + L_k < I_p - I_n$, i.e., when the net benefit to the government from strict regulation is less than the net benefit from imposing lax regulation, the government will choose lax regulation; When the public satisfies $H_s < H_c$, i.e., the damage caused by the public's direct belief in the reported content is less than the cost of making an objective identification, the public will choose to accept the reported content directly.

In Condition ②, the public will choose to believe the media reports directly regardless of the circumstances; an enterprise chooses the greenwashing strategy when it satisfies $Q_s - Q_h < Q_f - Q_k - Q_d$, i.e., the net benefit to the enterprise from making a green merger is less than the net benefit passed by making a greenwashing; The media will choose to report truthfully when they satisfy $M_s < M_c - Z_s$, the difference between the bribe gain from the media's collusion with the enterprise to report falsely and the gain from the loss of the enterprise's cooperation from the media's value-added reputation from reporting truthfully about the enterprise's content; The government will choose the strategy of lax regulation

**Table 3. Stability analysis of the equilibrium point of system $J$ in the four-party pure strategy solution.**

| Equilibrium point | Eigenvalues | Sign of the real part | Stability |
|---|---|---|---|
| $E_1(0,0,0,0)$ | $M_s - Q_f - Q_h + Q_k + Q_s,\ M_c - M_s - Z_s,\ A_c - A_s + I_n - I_p + L_k + L_s + R_c,\ H_s - H_c$ | $(x, x, x, x)$ | Stable under Condition ① |
| $E_2(0,1,0,0)$ | $Q_d - Q_f - Q_h + Q_k + Q_s,\ -M_c + M_s + Z_s,\ A_c - A_s + I_n - I_p + L_s + R_c,\ -H_c$ | $(x, x, x, -)$ | Stable under Condition ② |
| $E_3(0,0,1,0)$ | $M_s + L_s + Q_c - Q_f - Q_h + Q_k + Q_s,\ L_k + M_c - M_s - Z_s,\ -A_c + A_s - I_n + I_p - L_k - L_s - R_c,\ -H_c + H_s + \beta T_g$ | $(x, x, x, x)$ | Stable under Condition ③ |
| $E_4(0,1,1,0)$ | $L_s + Q_c + Q_d - Q_f - Q_h + Q_k + Q_s,\ M_s - M_c + Z_s - L_k,\ -A_c + A_s - I_n + I_p - L_s - R_c,\ \beta T_g - H_c$ | $(x, x, x, x)$ | Stable under Condition ④ |
| $E_5(1,0,0,0)$ | $Q_f - M_s + Q_h - Q_k - Q_s,\ M_c,\ A_c - A_s + I_n - I_p + L_k + R_c,\ H_s - H_c$ | $(x, +, x, x)$ | Unstable |
| $E_6(1,1,0,0)$ | $Q_f - Q_d + Q_h - Q_k - Q_s,\ -M_c,\ A_c - A_s + I_n - I_p + R_c,\ -H_c$ | $(x, -, x, -)$ | Stable under Condition ⑤ |
| $E_7(1,0,1,0)$ | $Q_f - Q_c - M_s - L_s + Q_h - Q_k - Q_s,\ L_k + M_c,\ A_s - A_c - I_n + I_p - L_k - R_c,\ H_s - H_c$ | $(x, +, x, x)$ | Unstable |
| $E_8(1,1,1,0)$ | $Q_f - Q_c - Q_d - L_s + Q_h - Q_k - Q_s,\ -L_k - M_c,\ A_s - A_c - I_n + I_p - R_c,\ -H_c$ | $(x, -, x, -)$ | Stable under Condition ⑥ |
| $E_9(0,0,0,1)$ | $M_s - Q_f - Q_h + Q_k + Q_s,\ L_c + M_c - M_s - Z_s,\ A_c - A_s + I_n - I_p + L_k + L_s + R_c,\ H_c - H_s$ | $(x, x, x, x)$ | Stable under Condition ⑦ |
| $E_{10}(0,1,0,1)$ | $Q_d - Q_f - Q_h + Q_k + Q_s,\ M_s - M_c - L_c + Z_s,\ A_c - A_s + I_n - I_p + L_s + R_c,\ H_c$ | $(x, x, x, +)$ | Unstable |
| $E_{11}(0,0,1,1)$ | $L_s + M_s + Q_c - Q_f - Q_h + Q_k + Q_s + \beta T_g,\ L_c + L_k + M_c - M_s - Z_s,\ A_s - A_c - I_n + I_p - L_k - L_s - R_c,\ -H_s + H_c - \beta T_g$ | $(x, x, x, x)$ | Stable under Condition ⑧ |
| $E_{12}(0,1,1,1)$ | $L_s + Q_c + Q_d - Q_f - Q_h + Q_k + Q_s + \beta T_g,\ M_s - L_k - M_c - L_c + Z_s,\ A_s - A_c - I_n + I_p - L_s - R_c,\ H_c - \beta T_g$ | $(x, x, x, x)$ | Stable under Condition ⑨ |
| $E_{13}(1,0,0,1)$ | $Q_f - M_s + Q_h - Q_k - Q_s,\ L_c + M_c,\ A_c - A_s + I_n - I_p + L_k + R_c,\ H_c - H_s$ | $(x, +, x, x)$ | Unstable |
| $E_{14}(1,1,0,1)$ | $Q_f - Q_d + Q_h - Q_k - Q_s,\ -L_c - M_c,\ A_c - A_s + I_n - I_p + R_c,\ H_c$ | $(x, -, x, +)$ | Unstable |
| $E_{15}(1,0,1,1)$ | $Q_f - M_s - Q_c - L_s + Q_h - Q_k - Q_s - \beta T_g,\ L_c + L_k + M_c,\ A_s - A_c - I_n + I_p - L_k - R_c,\ H_c - H_s$ | $(x, +, x, x)$ | Unstable |
| $E_{16}(1,1,1,1)$ | $Q_f - Q_c - Q_d - L_s + Q_h - Q_k - Q_s - \beta T_g,\ -L_c - L_k - M_c,\ A_s - A_c - I_n + I_p - R_c,\ H_c$ | $(x, -, x, +)$ | Unstable |

when it satisfies $A_c + R_c - A_s + L_s < I_p - I_n$,i.e., the net benefit to the government from lax regulation of the corporate media is greater than the benefit of strict regulation of it;.

In Condition ③, an enterprise will choose the greenwashing strategy when it satisfies $Q_s - Q_h + Q_c < Q_f - Q_k - M_s - L_s$, i.e., the net benefit that the enterprise obtains from conducting green acquisitions is less than the net benefit that it obtains from conducting greenwashing; When the media satisfy $M_c - Z_s < M_s - L_k$,i.e., the additional revenue gained by the media from truthful reporting of the enterprise's behavior is less than the additional revenue gained from false reporting, the media will choose to report the enterprise's behavior falsely; The government will choose the strategy of strict regulation when it satisfies,i.e., the net benefit to the government from lax regulation of the corporate media is less than the benefit from strict regulation of it; The public will choose the direct belief strategy when the public satisfies $H_s < H_c - \beta T_g$,i.e., when the public directly believes that the damage caused to itself by the media report is less than the damage caused by the objective identification.

In Condition ④, an enterprise chooses the greenwashing strategy when it satisfies $Q_s - Q_h + Q_c < Q_f - Q_k - Q_d - L_s$,i.e., the net benefit to the enterprise from making a green merger is less than the net benefit from making a greenwashing; The media will choose to report corporate behavior truthfully when they satisfy $M_s - L_k < M_c - Z_s$,i.e., the additional revenue gained by the media from reporting corporate behavior truthfully is greater than the additional revenue gained from reporting it falsely; The government will choose the strategy of strict regulation when it satisfies $I_p - I_n < A_c + R_c - A_s + L_s$, i.e., the net benefit to the government from lax regulation of the corporate media is less than the benefit from strict regulation of it; When the public satisfies $\beta T_g < H_c$,i.e., the cost to the public of objectively identifying the content of media reports is greater than the amount of corporate payout received as a result of the complaint, the public will choose the direct belief strategy.

In condition ⑤, the media will choose to report truthfully and the public will choose to believe it outright, regardless of the circumstances. Enterprises engage in greenwashing when they satisfy $Q_f - Q_k - Q_d < Q_s - Q_h$,i.e., when the enterprises gain from engaging in greenwashing is less than the gain from green acquisitions; The government will choose lax

regulation when it satisfies $A_c - A_s + R_c < I_p - I_n$,i.e., when the government's gain from strict regulation is less than the gain from lax regulation.

In Condition ⑥, the media chooses to report truthfully and the public chooses to believe what the media reports, regardless of the situation. Enterprises engage in greenwashing when they satisfy $Q_f - Q_k - Q_d - L_s < Q_s + Q_c - Q_h$,i.e., when the enterprises gain from engaging in greenwashing is less than the gain from green acquisitions; The government will choose to regulate strictly when it satisfies $I_p - I_n < A_c - A_s + R_c$,i.e., the government's gain from regulating strictly is greater than the gain from regulating loosely.

In condition ⑦, an enterprise will choose greenwashing when it satisfies $Q_s - Q_h < Q_f - Q_k - M_s$,i.e., the enterprise's gain from making a green merger is less than the gain from greenwashing; The media will choose to report falsely when they satisfy $M_c - Z_s < M_s - L_c$,i.e., when the additional revenue gained by the media from true reporting is less than the additional revenue gained from false reporting; When the government satisfies $A_c + R_c - A_s + L_s + L_k < I_p - I_n$, i.e., when the net benefit to the government from strict regulation is less than the net benefit from imposing lax regulation, the government will choose lax regulation; When the public satisfies $H_s > H_c$, i.e., when the damage caused by the public's direct belief in the content of the report is greater than the cost of making an objective identification, the public will choose to identify the content of the report objectively.

In condition⑧, an enterprise will choose greenwashing when it satisfies $Q_s - Q_h + Q_c < Q_f - Q_k - L_s - M_s - \beta T_g$, the enterprise's gain from making a green merger is less than the gain from greenwashing; The media will choose to report falsely when they satisfy $M_c - Z_s < M_s - L_c - L_k$,i.e., when the additional revenue gained by the media from true reporting is less than the additional revenue gained from false reporting; The government will choose the strategy of strict regulation when it satisfies $I_p - I_n < A_c + R_c - A_s + L_s + L_k$,i.e., the net benefit to the government from lax regulation of the corporate media is less than the benefit from strict regulation of it; and the public will choose the objective identification strategy when the public satisfies $H_s > H_c - \beta T_g$,i.e., when the public directly believes that the damage caused to itself by the media report is greater than the damage caused by the objective identification.

In Condition ⑨, enterprises choose greenwashing when they satisfy $Q_s - Q_h + Q_c < Q_f - Q_k - Q_d - L_s - \beta T_g$,i.e., when the benefits gained by the enterprise from making a green merger are less than the benefits gained from greenwashing; When the media satisfy $M_s - L_c - L_k < M_c - Z_s$,i.e., when the additional revenue gained by the media from false reporting is less than the additional revenue gained from truthful reporting, the media will choose to report truthfully about the enterprise's behavior; The government will choose to strictly regulate enterprises and the media when it satisfies $I_p - I_n < A_c - A_s + R_c + L_s$,i.e., the government's gain from lax regulation is less than the gain from strict regulation; The public will choose to objectively identify media reports when the public satisfies $H_c < \beta T_g$, i.e., the cost to the public of objectively identifying the content of the media reports is less than the amount of compensation that the whitewashing enterprise will pay to the public.

Through the above analysis of the stabilization point of the strategy this paper obtains the following conclusions:

(1) Conditions for determining the equilibrium point

By replicating the dynamic equations and analyzing the eigenvalues, this paper clarifies that the strategic choices of all parties, including enterprises, the media, the government and the public, will reach a stable state under specific conditions. For example, the equilibrium point of enterprises tends to favor green mergers and acquisitions (M&A) when the net benefit of enterprises' choosing green M&A exceeds greenwashing. Similarly, the strategies of the media, government and public are subject to similar conditional constraints.

(2) Equilibrium points under different strategy combinations

For enterprises, they will choose green M&A when the benefits of green M&A are greater than the benefits of greenwashing, and vice versa, this equilibrium is affected by the truthfulness of media reports and the strength of government regulation; for the media, they will choose truthful reporting when the benefits and reputation value-added of truthful

reporting are greater than the benefits of false reporting, and otherwise they will choose false reporting, and the media strategy directly affects the choices of enterprises and creates an equilibrium through its interaction with the government and the public; on the government side, the government chooses between strict regulation and lax regulation depending on the net benefits of the two, and chooses strict regulation when the social support and long-term benefits brought by strict regulation are higher than those by lax regulation. At the public level, the public will choose between rational identification or outright belief, depending on whether the benefits of rational identification outweigh its costs, and government regulation and media coverage will have a significant impact on the public's choice.

(3) Dynamic changes in the equilibrium point

The equilibrium point may shift in response to changes in external conditions as the strategic choices of the parties are influenced by each other. For example, the media's choice of false reports will prompt enterprises to favor greenwashing, while the strength of government regulation will also directly affect the location of the equilibrium point. By analyzing the eigenvalues of the equilibrium point, this paper clarifies the strategic stability of each party under different game conditions.

Based on the analysis of the above conclusions, this paper argues that in order to promote enterprises to actively carry out green M&A, the government should strengthen regulation, the media should insist on truthful reporting, and the public should improve the ability to rationally identify, and by coordinating the interactions of the four parties, the strategic choices of each party can be converged to the environmentally friendly green development path. In summary, the equilibrium point part reveals the interdependence of strategic choices of all parties, and the influence of external policies and market conditions on the stability of the equilibrium point, and effective policy intervention and regulatory measures can help to promote the parties of the main body of the game toward the equilibrium point of green and sustainable development.

## 5. Simulation of systems in different contexts

In order to verify the stability of the model while assigning values to different coefficients according to the real situation, and to consider the impact of their differences on the strategic choices of enterprises, media, government and public to explore the path of multi-constituents to jointly explore the green mergers and acquisitions of enterprises, this section carries out the systematic simulation in different scenarios through MATLAB. Specifically, it is worth thinking about how to propose governance countermeasures utilizing corporate and media considerations of their own reputations. Besides, as the victims of corporate greenwashing, it is worth exploring how the government and the public can be guided to adopt positive monitoring mechanisms and feedback mechanisms to govern corporate greenwashing.

In this study, we adopted a standardized approach by setting the annual revenue of a typical highly polluting enterprise at 100 units. This baseline allows us to express and compare the costs and benefits of various stakeholders—including enterprises, governments, media, and consumers—on a consistent scale. The main objective of this standardization is to facilitate comparability of economic effects across different stakeholders. Drawing on simulation-based designs from existing studies [49,50], we utilized data from recent policy documents, databases, references, news reports, and real-world cases as the primary sources.

According to studies on green mergers and acquisitions [30,31], the direct benefit for heavily polluting enterprises undertaking green M&A was set at 16 units, reflecting the positive impact of green M&A on corporate economic performance; The cost for heavily polluting enterprises to carry out green M&A is generally around 30% of their annual revenue, therefore, the cost was set at 30 units; Based on the China Carbon Neutrality Policy Documents and related research reports, we assigned a government subsidy of 4.5 units for green M&A by heavily polluting enterprises, equating to 15% of the M&A cost [16]; According to the Global Greenwashing Risk Report, greenwashing can yield short-term market gains of 15%–25% for heavily polluting enterprises. To account for the short-term market benefits of greenwashing, we assigned a

direct revenue of 15 units for such behavior; The Global Greenwashing Risk Report also states that greenwashing costs typically range from 5% to 10%. Combined with related research [34], we estimated the direct cost of greenwashing at 7 units, significantly lower than the 30 units required for green M&A, highlighting the relative affordability of greenwashing compared to substantive environmental actions; The Chongqing Municipal Ecological Environment Bureau reported that a major environmental incident involving Yao Hui Environmental Protection resulted in direct economic losses of nearly 10 million RMB, along with production interruptions and loss of market share. Drawing on this case and related studies [16,51], we assigned an exposure-induced loss of 8 units for enterprises engaged in greenwashing; According to an investigation by Caixin into the "21st Century Network Extortion Case" combined with data from China's environmental pollution surveys, the typical bribe paid by heavily polluting enterprises to media outlets is about 5%–8% of annual revenue. In our model, with annual revenue set at 100 units, we set the bribery cost at 6 units [20,52]; Using provisions from the *Water Pollution Prevention and Control Law of the People's Republic of China* (2017) and cases such as the Kinglong New Energy Vehicle Subsidy Fraud Case, the penalty for greenwashing enterprises was set at 9 units; From *the Annual Report on China's Media Industry* (2019) and related research [53], we found that increased media attention to corporate news is positively correlated with increased advertising revenue, which was standardized at approximately 10 units; Based on related research [19,54], the reputational gain from media truthfully exposing corporate behavior was set at 1 unit. According to the *Annual Report on China's Media Industry*, advertising revenue for mainstream media outlets decreased by 4%–8% year-over-year during periods of negative corporate news exposure. Therefore, we conservatively estimated that the commercial partnership loss due to truthful reporting was about 4 units; Using data from the *Notice from the General Administration of Press and Publication on Strengthening the Management of False News* (2021) and related studies [55], we set the total penalty cost for false reporting at 6 units; Drawing on data from notable false reporting cases published by Peking University Law and academic research [19,20,56], we set the reputational loss due to false reporting at 5 units; According to the *2023 China Environmental Statistics Yearbook and the 2020 Ministry of Ecology and Environment Report of China*, the comprehensive benefits derived from enhanced environmental regulation account for about 8% of regional economic indicators. Thus, we assigned direct benefits (excluding penalty revenue) from strict regulation at 8 units; Based on the Beijing Municipal Ecology and Environment Bureau's 2021 public environmental awareness survey and related research [28], we set the additional benefits for governments (e.g., reputation enhancement and public welfare improvement) from strict regulation at 1 unit; Using data from the 2023 China Environmental Statistics Yearbook, we found that expenditures on enforcement, monitoring, and related fields accounted for 7%–15% of total public spending on energy conservation and environmental protection. Combined with related research [57], we assigned strict regulatory costs at 7 units; According to related studies and official documents, relaxing environmental standards has yielded significant increases in tax revenue and employment, amounting to 13%–20% of annual corporate revenue. Thus, the basic revenue gained under lenient regulation was set at 15 units; The *2019 China Environmental Protest Incident Statistics Report* and related research [58] indicated that relaxed regulation could lead to societal losses equivalent to 2.97%. To simplify calculations, we set the loss under lenient regulation at 3 units; The World Bank estimated that environmental pollution causes economic losses amounting to 5.8% of China's GDP. If enterprises adopt green M&A and achieve cleaner production, this could reduce the public burden associated with pollution. Accordingly, we assigned a positive impact of 6 units for green M&A on the public; Data from the Institute of Public and Environmental Affairs (IPE) and reports such as the "A-Share Green Weekly" indicate that public access to accurate information has become easier. We set the average cost of public verification of corporate environmental behavior at 1 unit; The Beijing Municipal Ecology and Environment Bureau's 2021 public environmental awareness survey found that approximately 70% of respondents would file complaints when encountering environmental violations. However, in remote areas with less accessible regulatory channels and lower public awareness, complaint rates are as low as 10% [59]. Considering regional variations, we conservatively set the probability of public complaints against adverse corporate behavior at 40%; Based on the new Civil Code, related research, and the Chongqing Yao Hui Environmental Pollution Case, we calculated that compensation for public damages accounts

for 8.66% of annual revenue. Therefore, we set compensation to the public or society at 9 units; According to related research [60–62], consumers on average pay a premium of about 1.7% for environmentally friendly products. Considering economic and environmental factors, we quantified public losses due to misleading information at 2 units.

After processing the simulation data, the parameter settings in this paper are as follows:

$Q_s = 16$, $Q_h = 30$, $Q_c = 4.5$, $Q_f = 15$, $Q_k = 7$, $Q_d = 8$, $M_s = 6$, $L_s = 9$, $M_w = 10$, $M_c = 1$, $Z_s = 4$, $L_k = 6$, $L_c = 5$, $A_c = 8$, $R_c = 1$, $A_s = 7$, $I_p = 15$, $I_n = 3$, $G = 6$, $H_c = 1$, $\beta = 0.4$, $T_g = 9$, $H_s = 2$.

### 5.1 Corporate response mechanisms to compliance costs

Let the compliance cost $Q_h$ to be paid by enterprises for green M&A be $\{20, 30, 40\}$. The evolution process and results of the four-party game subjects are shown in the following figure:

According to the analysis in Fig 6, changes in the cost of green M&A significantly affect the strategy evolution trends of enterprises, the government and the public. Specifically, as the M&A cost increases, the probability of enterprises choosing green M&A gradually decreases, the probability of the government implementing strict regulation correspondingly increases, and the probability of the public objectively recognizing the content of media reports also increases. In the figure, this trend is shown as follows: while the M&A cost increases, the strategy evolution curve of enterprises decreases over time and eventually stabilizes at "0"; the strategy evolution curve of the government rises rapidly and eventually stabilizes at "1"; and the strategy evolution curve of the public increases from close to "0" to "1". Therefore, it can be inferred that appropriately reducing the cost of green M&A can help promote enterprises to choose green M&A strategies.

### 5.2 Government regulatory mechanisms

#### 5.2.1 Impact of changes in the Government's reputation. Let the reputation gain $R_c$ of strict government regulation be $\{3, 6, 9\}$, and the reputation loss $I_n$ of loose government regulation be $\{2.5, 3, 3.5\}$, the evolution process and results of the four-party game are shown in the figure below:

According to the analysis in Fig 7, the increase in the net value of the government's reputation due to regulation (i.e., the difference between the value added to the reputation due to strict regulation and the value subtracted from the reputation due to lax regulation) makes the government more inclined to choose strict regulation. This phenomenon is shown in the figure: as the net reputation value increases, the government's strategy evolution curve moves significantly faster towards "1" and eventually stabilizes at "1". When reputation net worth is low, although the government initially imposes strict regulation, with the multiple pressures brought by strict regulation (rising policy implementation costs, slowing economic growth, stakeholder opposition, etc.), the government may shift to lax regulation, resulting in a gradual tendency for

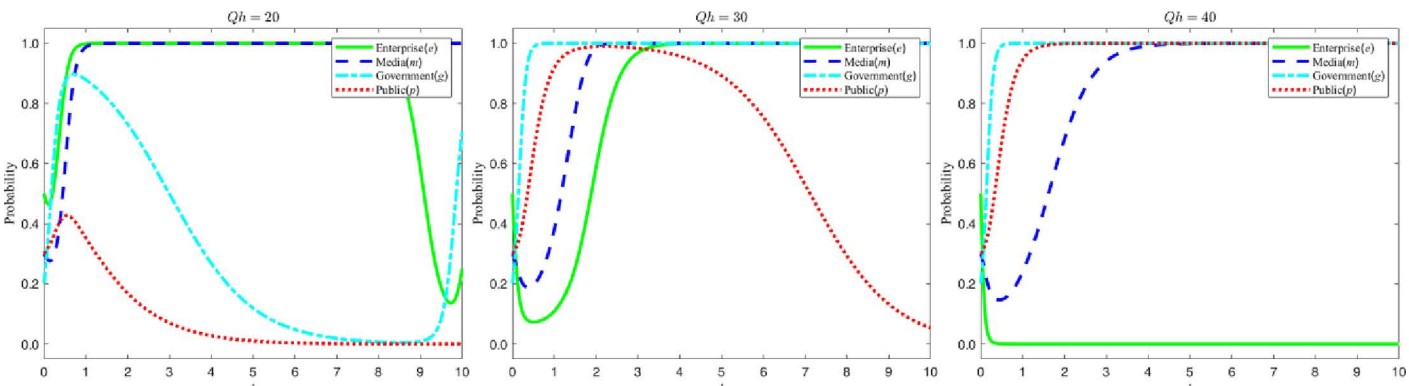

**Fig 6. Evolutionary curve trajectory at change.**

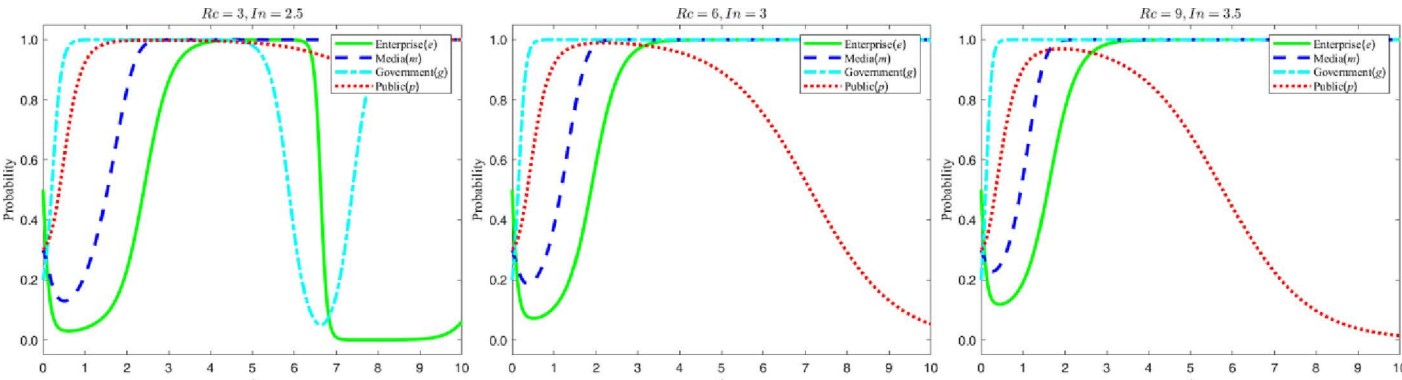

**Fig 7. Evolutionary curve trajectories when $R_c$ and $I_n$ change.**

enterprises to greenwash. However, in order to maintain long-term healthy socio-economic development, the government will re-enforce its regulatory efforts and shift from lax regulation to strict regulation to curb unfavorable corporate behavior. Therefore, attention needs to be paid not only to the respective reputational effects of strict and lax regulation, but also to the combined net effect produced by both. It is also worth noting that the strategic choices of the government as a single subject have a significant impact on other subjects, so the government should actively exercise its regulatory capacity to guide market behavior.

**5.2.2  Impact of changes in the level of government fines.**  Let the amount of fine $L_s$ by the government for greenwashing be $\{3, 9, 15\}$, and the amount of penalty $L_k$ by the government for false reporting by the media be $\{1, 6, 11\}$. The evolution process and results of the four-party game are shown in the figure below:

According to the analysis in Fig 8, changes in the amount of regulatory penalties imposed by the government on enterprises and the media not only affect the evolution of the government's own strategy, but also have a significant impact on the strategies of enterprises, the media, and the public. When the penalty amount rises, the government is more inclined to choose a strict regulatory strategy. Specifically, when the government raises the penalty for false media reporting, the media quickly shifts from false reporting to truthful reporting; when the penalty decreases, the public is more inclined to objectively identify the content of media reporting. This phenomenon is shown in the figure: as the penalty level rises, the media's strategy evolution curve shifts rapidly from stabilizing at "0" to "1". Similarly, when the government increases the

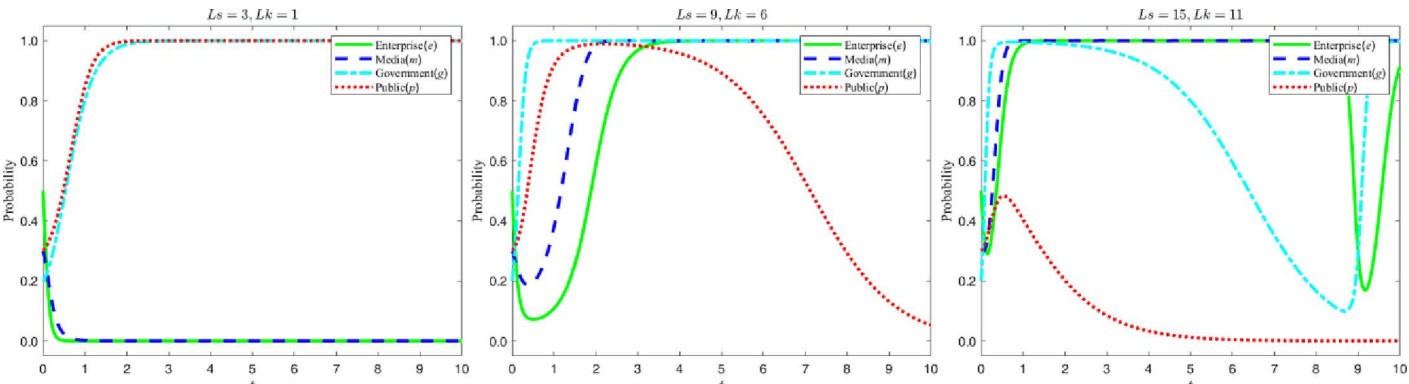

**Fig 8. Evolutionary curve trajectories when $L_s$ and $L_k$ change.**

penalty for greenwashing, the probability of enterprises choosing greenwashing decreases significantly, and the lowest point of the strategy evolution curve of enterprises in the figure increases and the time needed to stabilize at "1" is shortened. At the same time, as the amount of punishment rises, the public's strategy evolution curve gradually changes from objective identification to direct belief in the media, and finally stabilizes at "0".

It is worth noting that when enterprises choose "green M&A" for a long time, and the media continues to choose "truthful reporting", the government will gradually shift to a lax regulatory strategy due to the pressure of strict regulation (e.g., higher costs of policy implementation, increased social supervision, resource consumption, etc.). The government will gradually shift to a more lenient regulatory strategy. Therefore, it can be concluded that a moderate level of penalty can effectively guide the behavior of enterprises and the media, but at the same time, the government also needs to balance the relationship between regulatory costs and long-term social benefits.

### 5.3 Impact of changes in the amount of bribes paid to the media

Let the amount of bribe $M_s$ received by the media from the enterprise be $\{3, 6, 9\}$, and the evolution process and results of the four-party game subjects are shown in the figure below:

According to the analysis in Fig 9, as the amount of bribes paid by enterprises to the media increases, the probability of the media choosing false reports significantly increases and the time required to shift from false reports to true reports is extended. However, the higher cost of bribery makes enterprises become more cautious in choosing greenwashing strategies. In the figure, this trend is shown by the fact that the lowest point of the media's strategy evolution curve decreases with the increase of the bribe amount, while the lowest point of the enterprise's strategy evolution curve rises with the increase of the bribe amount, and the time required for the enterprise's strategy curve to stabilize at "1" is significantly shortened.

It follows that although bribery can increase enterprises' utilization of false reporting in the short run, the high cost of bribery forces enterprises to consider greenwashing strategies more prudently in the long run. In addition, bribery can also exacerbate the difficulty of choosing between false and true reporting in the media, slowing their transition to true reporting.

### 5.4 Mechanisms for public influence

**5.4.1 The effect of high and low identification costs on the public's choice of strategy.** Let the cost $H_c$ for the public to objectively recognize the content of media reports be $\{0.5, 1, 1.5\}$. The evolution of the four-party game players and the results are shown in the figure below:

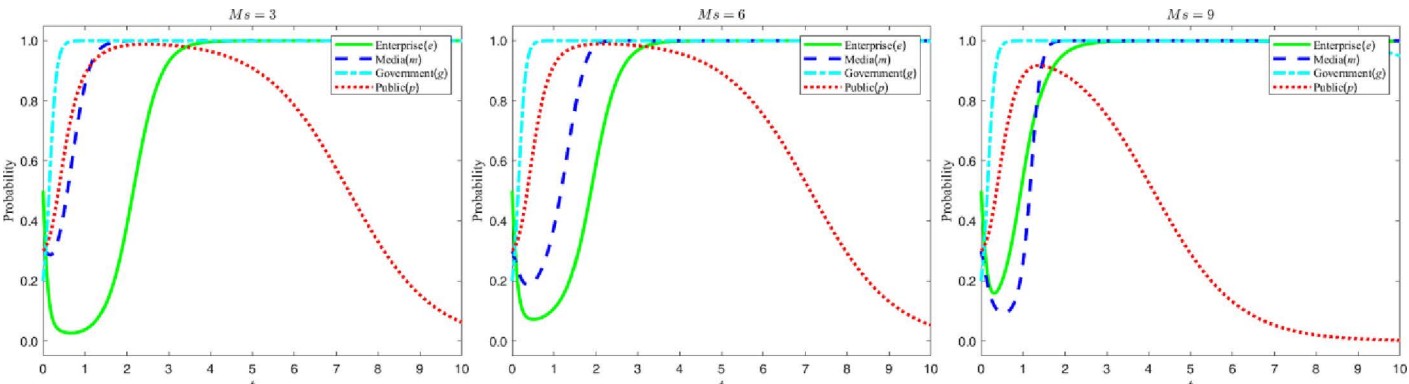

**Fig 9. Evolutionary curve trajectories at a change in.** $M_s$.

According to the analysis in Fig 10, when the cost for the public to objectively identify the content of media reports increases, the public is more inclined to directly believe the media reports instead of making independent judgments. In the figure, this trend is shown as follows: as the cost of identification increases, the time for the public's strategy evolution curve to stabilize at "1" becomes shorter and shorter, and the speed of transformation to "0" accelerates significantly. This shows that increasing the public's cost of objectively identifying media reports will weaken their independent judgment and make them more receptive to media reports. It also suggests that lowering the public's identification cost may be an effective means of enhancing their information screening ability, so as to maintain rational judgment in information dissemination.

**5.4.2 Impact of high and low complaint rates on enterprises' strategy choices.** Let the public complaint rate $\beta$ be $\{0, 0.4, 0.8\}$, and the evolution process and results of the four-party game subjects are shown below:

According to the analysis in Fig 11, when the public complaint rate rises, enterprises will more actively choose green M&A strategies, and the media will therefore be more inclined to report truthfully. In the figure, this trend is shown as follows: as the complaint rate increases, the strategy evolution curve of enterprises rapidly shifts from stabilizing at "0" to stabilizing at "1"; at the same time, the media strategy evolution curve also gradually shifts from fluctuating within the selection interval to stabilizing at "1". This shows that the public's complaint behavior plays a positive guiding role in promoting the enterprises to choose green M&A and the media to report truthfully, which indicates that enhancing the public's supervision can help to promote the enterprises' green transformation and the media's truthful reporting.

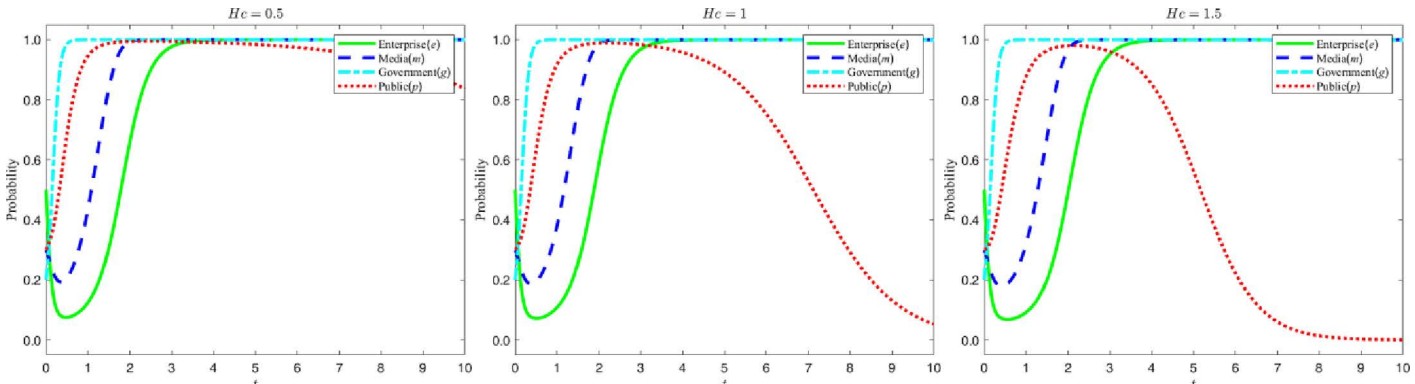

**Fig 10. Evolutionary curve trajectories at a change in.** $H_c$.

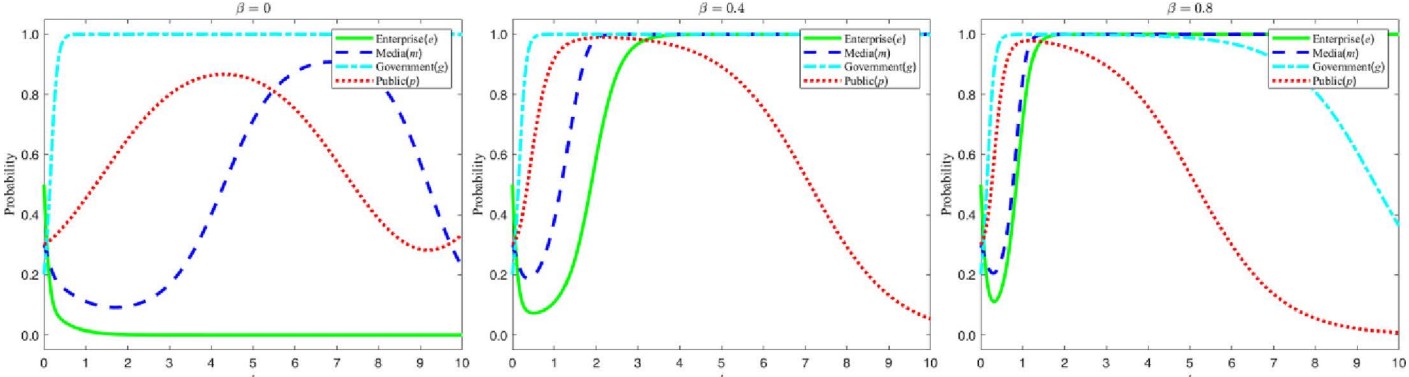

**Fig 11. Evolutionary curve trajectories at a change in.** $\beta$.

 

# 6. Discussion

## 6.1 Innovations

This paper incorporates the strategic interactions among heavily polluting enterprises, government, media, and the public within a unified game framework, systematically revealing the strategy evolution paths of these parties under different interest drivers. Simulation results show that, through collaboration among government, media, and the public, the likelihood of heavily polluting enterprises opting for green mergers and acquisitions significantly increases. This innovation not only fills a gap in multi-party interaction analysis within green M&A research but also provides an extensible analytical model for future studies, validating that multi-stakeholder participation and cooperation can effectively guide corporate behavior.

Additionally, the model innovatively includes the bribery of media by enterprises, showing through simulations the influence of bribe amounts on media's truthful reporting and corporate greenwashing choices. Results indicate that when bribery costs are prohibitively high, heavily polluting enterprises tend to choose green M&A in the long term, while the media increasingly favors truthful reporting. This finding suggests that raising bribery costs or strengthening anti-bribery regulations can curb corporate manipulation of media, offering a new pathway for managing greenwashing.

Furthermore, the simulation analysis reveals the synergistic effects of government regulation, media oversight, and public participation in promoting green M&A. Specifically, strict government penalties, truthful media reporting, and the public's rational discernment work together to significantly suppress greenwashing. By comparing different policy combinations, this paper confirms an optimal path for enhancing collaborative mechanisms: adjusting regulatory intensity, increasing media independence and transparency, and reducing the public's information discernment costs. This innovation provides theoretical support for policy formulation through multi-party coordination.

Through dynamic evolution analysis, the paper explores the evolution and stability of each party's strategies under different policy and market conditions. Simulation results demonstrate that adjustments in government regulation intensity are particularly critical in influencing the strategic choices of both heavily polluting enterprises and the media. Increasing government fines and incentives gradually raises the proportion of green M&A, while lenient regulation leads enterprises to favor greenwashing. This finding validates the dynamic impact of policy adjustments, supporting flexible and adaptive green transition policies and highlighting the importance of dynamic policy adjustment in green governance for heavily polluting enterprises.

Based on simulation analysis, this paper offers strategic optimizations and policy recommendations for a multi-stakeholder collaboration mechanism. By modeling various scenarios, it systematically analyzes the impact of government subsidies, punitive measures, media independence, and public oversight on the green M&A choices of heavily polluting enterprises. The recommendations include reducing green M&A costs, increasing policy incentives, enhancing media transparency, and using public reporting and feedback mechanisms to improve corporate environmental transparency. Simulation results show that, under multi-party influence, these measures effectively increase the appeal of green M&A and reduce greenwashing occurrences, providing policymakers with practical policy tools.

## 6.2 Answers to the questions

Q1: How can a multi-party game mechanism reduce the occurrence of greenwashing behaviors by heavily polluting enterprises and encourage them to proactively choose green mergers and acquisitions?

A multi-party game mechanism can effectively reduce the probability of greenwashing and encourage green M&A among heavily polluting enterprises. Simulation analysis shows that truthful media reporting and rational public discernment play a significant role in suppressing greenwashing. Specifically, when the media opts for truthful reporting rather than false narratives, enterprises are more likely to choose green M&A due to concerns over negative exposure. Additionally, lowering the public's cost of identifying false information—such as by providing more transparent environmental

information—enhances the public's ability to distinguish genuine environmental practices. These collaborative mechanisms guide enterprises toward authentic green transformations rather than superficial greenwashing.

Q2: Under a combination of government incentives and constraints, how can the economic benefits of green mergers and acquisitions be enhanced to make them the priority choice for enterprises?

To make green M&A the preferred choice for heavily polluting enterprises, government incentives and constraints should focus on increasing the economic appeal of green M&A. Simulation analysis indicates that providing subsidies and tax benefits for mergers enables enterprises to gain higher economic returns from green M&A, thereby strengthening their motivation to choose this path. Additionally, when the government imposes high fines and strict penalties for greenwashing, enterprises are more likely to choose green M&A to avoid the costs of non-compliance. By combining subsidies with penalties, the government can significantly enhance the economic benefits of green M&A, making it the preferred strategy for environmental transformation.

Q3: How can the supervisory mechanisms of media and the public be leveraged to increase transparency in corporate environmental practices, effectively identifying and curbing greenwashing behaviors?

This study shows that media and public supervision play a critical role in curbing greenwashing and enhancing transparency in corporate environmental practices. Simulation analysis reveals that when the media reports truthfully, enterprises tend to choose green M&A to avoid negative consequences. Moreover, the public can provide timely feedback on corporate misconduct through reasonable supervision and reporting mechanisms, further increasing transparency. Reducing the public's identification costs—such as by strengthening environmental education and providing accessible environmental information—also helps the public more accurately assess corporate environmental actions, thereby reducing greenwashing. Supervision by media and the public forms an effective constraint on enterprises, ensuring the authenticity and reliability of their environmental practices.

## 6.3 Research limitations

This paper also has some limitations:

(1) Model Simplification Issue: This study's game model assumes bounded rationality among the parties to simplify analysis and computation. However, in reality, the decision-making processes of heavily polluting enterprises, government, media, and the public are often more complex and influenced by various uncertainties. While the simplified model captures the core interactions of multi-party games, it does not fully reflect the decision-making details and complexities present in real environments. Therefore, the model results may partially deviate from actual situations.

(2) Data Support Limitations: The simulation analysis in this study primarily uses data from industry reports and public databases, which have limitations in terms of availability and accuracy, especially regarding the economic impacts of green M&A and greenwashing. A lack of comprehensive empirical data limits the ability to validate model parameters effectively. Future research could incorporate more empirical data to improve the reliability of model parameters and the persuasiveness of conclusions.

(3) Neglect of External Factors: This study focuses mainly on the game interactions among heavily polluting enterprises, government, media, and the public, without adequately considering the potential impact of external economic conditions and international policy changes on green M&A and greenwashing. For example, global economic fluctuations and shifts in international environmental policies may significantly influence the environmental decisions of heavily polluting enterprises. Future studies could integrate external factors into the model to enhance its real-world applicability.

(4) Neglect of Policy Heterogeneity: Government policies on environmental regulation and incentives vary significantly across regions, yet this study does not distinguish between regional policy differences. In practice, regional policy

measures may lead to different behaviors among heavily polluting enterprises regarding green M&A and greenwashing. Future research could incorporate policy heterogeneity into the model to further analyze game outcomes under different policy environments.

## 7. Conclusions

This paper constructs a four-party game model to analyze the strategic choices of heavily polluting enterprises between green mergers and acquisitions and greenwashing. It reveals the critical role of interactions among the government, media, and public in driving the green transition of these enterprises. The results indicate that strengthening multi-stakeholder collaboration mechanisms not only guides heavily polluting enterprises toward green M&A but also provides new approaches for balancing environmental and economic benefits.

Firstly, the paper confirms the counterbalancing effect of multiple stakeholders in the green transition. Genuine media oversight and rational public engagement can effectively increase transparency in heavily polluting enterprises, curbing greenwashing. The government, through appropriate incentives and penalties, offers these enterprises more attractive green M&A options. These effects interlink, creating a multi-layered green governance structure.

Additionally, simulation analysis shows that optimizing policy design and enhancing multi-party cooperation can establish a dynamic, stable path toward long-term green development. The collaboration of multiple stakeholders and the flexible adjustment of policies help heavily polluting enterprises gradually achieve green transformation under external pressure and incentives, adding momentum to sustainable industry development.

In summary, this study provides theoretical support and practical insights for multi-stakeholder collaborative governance in green transition. Future research could incorporate more empirical data and complex environmental factors to further refine and expand the model, providing a more scientific foundation for green governance policy-making.

## Author contributions

**Conceptualization:** Mengxin Sun.

**Data curation:** Mengxin Sun, Xianggang Huang.

**Formal analysis:** Mengxin Sun.

**Resources:** Mengxin Sun, Xianggang Huang.

**Writing – original draft:** Mengxin Sun.

**Writing – review & editing:** Xianggang Huang.

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
