## [Decision Letter · Decision Letter 0]

30 Jan 2025

PONE-D-25-00481Cooperative pathways for the green transformation of heavily polluting enterprises: a four-party game-driven mechanism for green M&A

PLOS One

Dear Dr. sun,

Thank you for submitting your manuscript to PLOS ONE. After careful consideration, we feel that it has merit but does not fully meet PLOS ONE’s publication criteria as it currently stands. Therefore, we invite you to submit a revised version of the manuscript that addresses the points raised during the review process.

Comments from the Academic Editor: We note that you have derived simulation parameters (Section 5, Pages 49–50) from industry reports and databases; Green M&A metrics came from the China Green Development Report and Annual Report on Global Renewable Energy Markets. Subsidies were based on the China Carbon Neutral Policy Paper, and greenwashing data from the Global Greenwashing Risk Report. Additional parameters used the Cathay Pacific Database.

However, in order to comply with PLOS One's criteria 3 for publication which states that experiments and analyses must be conducted rigorously, with appropriate controls; methods must be described in sufficient detail for others to replicate the analyses; (http://journals.plos.org/plosone/s/criteria-for-publication), you will need to add explicit citations, include URLs/DOIs, and expand the Methods to clearly explain and describe the source of each of the parameter values used for the simulation analysis. Adding this data compiled as a supplementary material is another recommendation to consider.

Comments from PLOS Editorial Office: We note that one or more reviewers has recommended that you cite specific previously published works. As always, we recommend that you please review and evaluate the requested works to determine whether they are relevant and should be cited. It is not a requirement to cite these works. We appreciate your attention to this request.

We look forward to receiving your revised manuscript.

Kind regards,

Manuel Herrador, Ph.D.

Academic Editor

PLOS ONE

Additional Editor Comments:

Dear authors,

I have reviewed your manuscript and kindly request minor corrections suggested by one reviewer.

Best regards

Reviewers' comments:

Reviewer's Responses to Questions

**Comments to the Author**

1. Is the manuscript technically sound, and do the data support the conclusions?

Reviewer #1: Yes

Reviewer #2: Yes

2. Has the statistical analysis been performed appropriately and rigorously? 

Reviewer #1: Yes

Reviewer #2: Yes

3. Have the authors made all data underlying the findings in their manuscript fully available?

Reviewer #1: Yes

Reviewer #2: Yes

4. Is the manuscript presented in an intelligible fashion and written in standard English?

Reviewer #1: Yes

Reviewer #2: Yes

5. Review Comments to the Author

Reviewer #1: 1、This paper explores the behavioral game theory between four stakeholders: government, media, heavily polluting enterprises, and the public. It is a relatively rigorous management science paper.

2、The introduction section can be expanded to include the research motivations and potential marginal contributions, and a comparison with previous studies should be analyzed and discussed.

3、What exactly does "the cost of strict regulation for the government" and "the basic revenue gained by the government from lenient regulation" refer to?

4、While it is understandable for enterprises to accept regulation, why would "the media" also "accept regulation"? The logic in Figure 1 should be reconsidered—why would the media accept regulation?

5、Can the decision-making motives and influencing mechanisms of each stakeholder be further clarified to help readers better understand the interactions among stakeholders in the game model?

6、The mechanism description in this paper is very rigorous, but there is a small flaw. The author is recommended to refer to https://doi.org/10.1016/j.esr.2024.101590 and https://doi.org/10.3390/agriculture15010023 to further enhance the content.

7、Are the parameter values in the numerical analysis based on references from literature or real-world cases?

8、Is this game model applicable only to China, or can it also be effective in other countries or regions?

Reviewer #2: You have done excellent work in the field of Game theory, especially in terms of the four agents. Generally, revision is necessary for the manuscript, however, this paper can be accepted now due to its high quality.

6. PLOS authors have the option to publish the peer review history of their article (what does this mean? ). If published, this will include your full peer review and any attached files.

**Do you want your identity to be public for this peer review?** For information about this choice, including consent withdrawal, please see our Privacy Policy .

Reviewer #1: No

Reviewer #2: No

---

## [Author Response · Author response to Decision Letter 1]

12 Mar 2025

Dear Editors and Reviewers,

We sincerely appreciate your valuable feedback and suggestions for improving our manuscript. We fully accept your recommendations and have revised the manuscript accordingly based on your comments. Please refer to the uploaded files for detailed modifications.

Best regards,

Authors

---

## [Decision Letter · Decision Letter 1]

21 Mar 2025

Cooperative pathways for the green transformation of heavily polluting enterprises: a four-party game-driven mechanism for green M&A

PONE-D-25-00481R1

Dear Dr. sun,

We’re pleased to inform you that your manuscript has been judged scientifically suitable for publication and will be formally accepted for publication once it meets all outstanding technical requirements.

Kind regards,

Manuel Herrador, Ph.D.

Academic Editor

PLOS ONE

Additional Editor Comments (optional):

Dear authors,

I am pleased to inform you that your paper has been accepted for publication in PLOS One.

Thank you for your valuable contribution.

Best regards

Reviewers' comments:

Reviewer's Responses to Questions

**Comments to the Author**

1. If the authors have adequately addressed your comments raised in a previous round of review and you feel that this manuscript is now acceptable for publication, you may indicate that here to bypass the “Comments to the Author” section, enter your conflict of interest statement in the “Confidential to Editor” section, and submit your "Accept" recommendation.

Reviewer #1: All comments have been addressed

2. Is the manuscript technically sound, and do the data support the conclusions?

Reviewer #1: Yes

3. Has the statistical analysis been performed appropriately and rigorously? 

Reviewer #1: Yes

4. Have the authors made all data underlying the findings in their manuscript fully available?

Reviewer #1: Yes

5. Is the manuscript presented in an intelligible fashion and written in standard English?

Reviewer #1: Yes

6. Review Comments to the Author

Reviewer #1: The article has been thoroughly revised and is now very well-polished. The authors have addressed the previous comments in a comprehensive and thoughtful manner, significantly improving the overall quality of the manuscript. The structure is clear, the arguments are well-articulated, and the methodology is sound and appropriate for the research questions posed. Furthermore, the findings are relevant and contribute meaningfully to the existing literature. The writing is coherent and professional, and the paper meets the standards expected for publication. Given the substantial improvements and the academic value of the study, I recommend the article for acceptance without further revision.

7. PLOS authors have the option to publish the peer review history of their article (what does this mean? ). If published, this will include your full peer review and any attached files.

**Do you want your identity to be public for this peer review?** For information about this choice, including consent withdrawal, please see our Privacy Policy .

Reviewer #1: No

---

## [Editor Report · Acceptance letter]

PONE-D-25-00481R1

PLOS ONE

Dear Dr. Sun,

I'm pleased to inform you that your manuscript has been deemed suitable for publication in PLOS ONE. Congratulations! Your manuscript is now being handed over to our production team.

Kind regards,

on behalf of

Dr. Manuel Herrador

Academic Editor

PLOS ONE